# Dispersive Fourier transform based dual-comb ranging

Bing Chang[1,9], Teng Tan [1,2,3,9], Junting Du[1,9], Xinyue He[1], Yupei Liang[1], Zihan Liu[1], Chun Wang[1,4], Handing Xia[4], Zhaohui Wu[4], Jindong Wang [5], Kenneth K. Y. Wong [6], Tao Zhu[5], Lingjiang Kong[2], Bowen Li [1] ✉, Yunjiang Rao [1,7] ✉ & Baicheng Yao [1,8] ✉

Laser-based light detection and ranging (LIDAR) offers a powerful tool to real-timely map spatial information with exceptional accuracy and owns various applications ranging from industrial manufacturing, and remote sensing, to airborne and in-vehicle missions. Over the past two decades, the rapid advancements of optical frequency combs have ushered in a new era for LIDAR, promoting measurement precision to quantum noise limited level. For comb LIDAR systems, to further improve the comprehensive performances and reconcile inherent conflicts between speed, accuracy, and ambiguity range, innovative demodulation strategies become crucial. Here we report a dispersive Fourier transform (DFT) based LIDAR method utilizing phase-locked Vernier dual soliton laser combs. We demonstrate that after in-line pulse stretching, the delay of the flying pulses can be identified via the DFT-based spectral interferometry instead of temporal interferometry or pulse reconstruction. This enables absolute distance measurements with precision starting from 262 nm in single shot, to 2.8 nm after averaging 1.5 ms, in a non-ambiguity range over 1.7 km. Furthermore, our DFT-based LIDAR method distinctly demonstrates an ability to completely eliminate dead zones. Such an integration of frequency-resolved ultrafast analysis and dual-comb ranging technology may pave a way for the design of future LIDAR systems.

Originating from the physical principle of measuring the optical interference[1] or the time of flight (TOF)[2], laser-based light detection and ranging (LIDAR)[3] has become a critical technology in industrial and scientific metrology. It offers high-precision, extensive range, and fast acquisition[4,5]. Over recent decades, the advent of the optical frequency comb has delivered unrivaled rulers for ultra-precise time-frequency measurements[6]. Specifically, Frequency combs intrinsically generate highly coherent pulse sequences, yielding revolutionary light sources for absolute distance and displacement detection[7–9]. Recently, the development of comb stabilization technology[10,11], further enables ultrahigh accuracy for LIDARs[12–15]. Besides, rapid progress of micro-comb technology[16,17] opens a way to miniaturized optical ranging[18,19].

[1]Key Laboratory of Optical Fiber Sensing and Communications (Education Ministry of China), University of Electronic Science and Technology of China, Chengdu 611731, China. [2]School of Information and Communication Engineering, University of Electronic Science and Technology of China, Chengdu 611731, China. [3]Institute of Electronic and Information Engineering of UESTC, Guangdong 523808, China. [4]Research Center of Laser Fusion, China Academic of Engineering Physics, Mianyang 621900, China. [5]Key Laboratory of Optoelectronic Technology & Systems (Education Ministry of China), Chongqing University, Chongqing 400044, China. [6]Department of Electrical and Electronic Engineering, University of Hong Kong, Pokfulam Road, Hong Kong SAR 990777, China. [7]Research Centre for Optical Fiber Sensing, Zhejiang Laboratory, Hangzhou 310000, China. [8]Engineering Center of Integrated Optoelectronic & Radio Meta-chips, University of Electronic Science and Technology, Chengdu 611731, China. [9]These authors contributed equally: Bing Chang, Teng Tan, Junting Du. ✉e-mail: bowen.li@uestc.edu.cn; yjrao@uestc.edu.cn; yaobaicheng@uestc.edu.cn

Microcombs with repetition rates tens to hundreds gigahertz[20,21], enable the acquisition rate higher than 100 MHz[22]. Moreover, the frequency channels of a microcomb can be conveniently filtered out as frequency modulation continuous waves[23–25] or chaotic waves[26,27], further support massively parallel ranging. In addition to using one single comb, for taking full advantage of the comb spectra meanwhile breaking the bandwidth limits of optical spectrometers, dual-comb-based systems with high-resolution time-resolved interferograms[28–30], demonstrate the capability to combine both high accuracy and large non-ambiguity range (NAR) for LIDARs[31–33]. In many cases, the phase-locked dual-comb can even be generated in one cavity (either a fiber-loop or a microresonator)[34–38], further suppressing the common noise and simplifying the locking system. However, the performance of a dual-comb ranging system is not only contingent on the uncertainty of the light sources but also heavily influenced by the resolution of the signal demodulation strategy[39]. The conventional TOF schemes, based on the identification of pulse overlaps, commonly present an inherent conflict: increasing the dual-comb repetition difference enhances the measurement speed but at the cost of reduced pulse fitting accuracy. Moreover, in conventional dual-comb ranging techniques, it is often challenging to circumvent dead-zones[40,41].

Here, to the best of our knowledge, for the first time we introduce the dispersive Fourier transform (DFT) concept into a dual-comb ranging system and achieve the stretched pulse-to-pulse spectrally interferometric analysis for the TOF measurements. This scheme enables absolute distance measurement via one-single-shot acquisition without the need of repeatedly sampling the pulse train[42–44]. Since the accuracy of the DFT is irrelevant to the parametric design of frequency comb, the achieved performance features both high detection speed and extreme precision. Experimental results demonstrate that we can achieve 262 nm ranging accuracy in single shot, and the ranging accuracy reaches 2.8 nm after averaging 1.5 ms. Leveraging the dual-comb-based Vernier effect[33,45], the NAR can be extended to 1.7 km. That suggests the distance-precision ratio is approaching $10^{12}$. Moreover, since this method uniquely does not require solving pulse envelopes, the probe-reference pulse offset can be precisely obtained when they are very close, thus completely eliminate dead-zones[46]. We successfully use this system in high-precision morphology measurement and real-time detection of fast dynamic motion. Keeping the high scalability of the dual-comb Vernier, such a combination of ranging and DFT offers unique advances. By measuring intra-pulse interference rather than temporal separation, this approach surpasses the sampling limitation in conventional dual-comb rangefinders and thus overcomes the trade-off between speed and accuracy. It may provide a new platform for broad applications ranging from positioning, imaging, to deformation analysis.

## Results

Figure 1a illustrates the conceptual design of our DFT-based dual-comb vernier rangefinder. To measure the TOF of optical pulses, we utilize two phase-locked soliton fiber laser combs, one as the signal comb and the other as the local comb. Among them, the signal comb is divided into two paths, one is used as a probe, and the other as a reference. At the receiving terminal, we analyze the synthesized signal carrying ranging information through the DFT. The two laser combs have slightly different repetition rates $\Delta f_r$, which enables us to realize a maximum $L_{NAR} = c/2\Delta f_r$, by conveniently exchanging the roles of the two combs with repetition rates $f_{r1}$ and $f_{r2}$[18], here $c$ is the light velocity. Figure 1b shows the operation principle. When using two frequency combs with distinct repetition periods $T_1$ and $T_2$, the TOF $\Delta\tau$ is enlarged to be $\Delta t = m\Delta\tau$, here $m = f_{r1}/\Delta f_r$, also known as the zooming factor. In this operation, the sampling step $\Delta T$ equals to $|1/f_{r1} - 1/f_{r2}| = \Delta f_r/f_{r1}f_{r2}$, while the sampling period $T_{update}$ is $1/\Delta f_r$. The measured distance $L = c\Delta\tau/2$. In conventional dual-comb ranging methods, the $\Delta\tau$ was typically estimated via fitting the interferometric pulse envelope. In

this case, for meeting the Nyquist sampling theorem, there is an inherent limitation $\Delta f_r < f_{r1}f_{r2}/2B$, here $B$ is the 3-dB spectral bandwidth of the probe comb. This is a main reason that typical fiber dual-comb system cannot measure rapidly. On the other hand, in practice, $f_{r1}$ and $f_{r2}$ are not necessarily integer multiples of their repetition frequency difference (i.e., $m$ is not an integer), thus their pulses are difficult to be strictly aligned in the time domain. Specifically, pulse-to-pulse offsets between the reference comb and the local comb could be $\tau_1$ (between pulse i and iii) and $\tau_2$ (between pulse ii and iv), while pulse-to-pulse offsets between the probe comb and the local comb could be $\tau_3$ (between pulse v and vii) and $\tau_4$ (between pulse vi and viii), here $\tau_1 + \tau_2 = \tau_3 + \tau_4 = \Delta T$.

Now we explain our strategy. The left green box shows the region that the reference pulses overlap with the local pulses, while the right yellow box shows the region that the returned probe pulses (with a TOF delay) overlap with the local pulses, the accurate TOF $\Delta\tau = T_1 - N(T_1 - T_2) + \tau_1 - \tau_3$, or $\Delta\tau = T_2 - N(T_1 - T_2) + \tau_2 - \tau_4$. Therefore, the distance to be tested can be expressed as:

$$L = \frac{(T_1 - N\Delta T + \tau_1 - \tau_3)c}{2} = \frac{(T_2 - N\Delta T + \tau_2 - \tau_4)c}{2} \quad (1)$$

In a conventional comb ranging system, to identify $\tau_1$ to $\tau_4$ is not easy, as commonly these numbers are too small (e.g., down to fs level) to well detect. In our design, taking advantage of the DFT, we can detect the $\tau_1$ to $\tau_4$ with high resolution via intra-pulse interference. The principle of the DFT is shown in Fig. 1c schematically. According to the space-time duality, the demonstration of spectral information in time domain can be analogized to the spatial Fourier transform based on Fraunhofer's diffraction[42]. Such a large group velocity dispersion (GVD) adds a linear frequency chirp and broadens the pulses. A broadened pulse can have an enlarged temporal width $\delta$,

$$\delta = \left| \frac{-2\pi c \beta_2}{\lambda^2} \right| L\Delta\lambda \quad (2)$$

Here $\beta_2$ is the GVD parameter, $\lambda$ is the central wavelength of the comb pulses, $L$ is the length of dispersive element, and $\Delta\lambda$ is the 3-dB spectral bandwidth of the pulse. Such pulse broadening is also particularly advantageous in ranging applications, especially for enhancing pulse energy through the chirped-pulse amplification technique[47]. More results are shown in Supplementary Note S3.1 (Fig. S7). After DFT, the temporal shape of the broadened pulse is the same as its original spectral shape. When two pulses with a small time delay are simultaneously transmitting in a dispersive element, the two pulses will be co-stretched and overlap each other in a photodetector. The frequency of their interferometric fringes is given by:

$$f_i = \frac{\tau}{2\pi|\beta_2|L} \quad (3)$$

Here $\tau$ is the delay of the two pulses. As a result, we can obtain the relative delay of the two pulses through the spectral analysis of the interference fringes. To satisfy the far-field diffraction condition[48], the $|\beta_2|L$ must be larger than $\tau_0^2/(2\pi)$. Here $\tau_0 = 368$ fs is the original pulse duration.

Figure 1d demonstrates our experimental setup. The dual-comb source consists of two stabilized mode-locked fiber laser combs. We use fiber laser combs because their cavity parameters are flexibly editable. The comb 1 and comb 2 respectively have repetition rates of 24.465 MHz and 24.55 MHz, offering a $\Delta f_r = 85$ kHz. They are phase-locked on two radio-frequency clocks through two digital feedback loops. After averaging, the minimum uncertainty of the dual-comb repetition rates is smaller than 0.25 mHz. More characterizations of the dual-comb source are shown in Supplementary Note S2 (Figs. S4, S5).

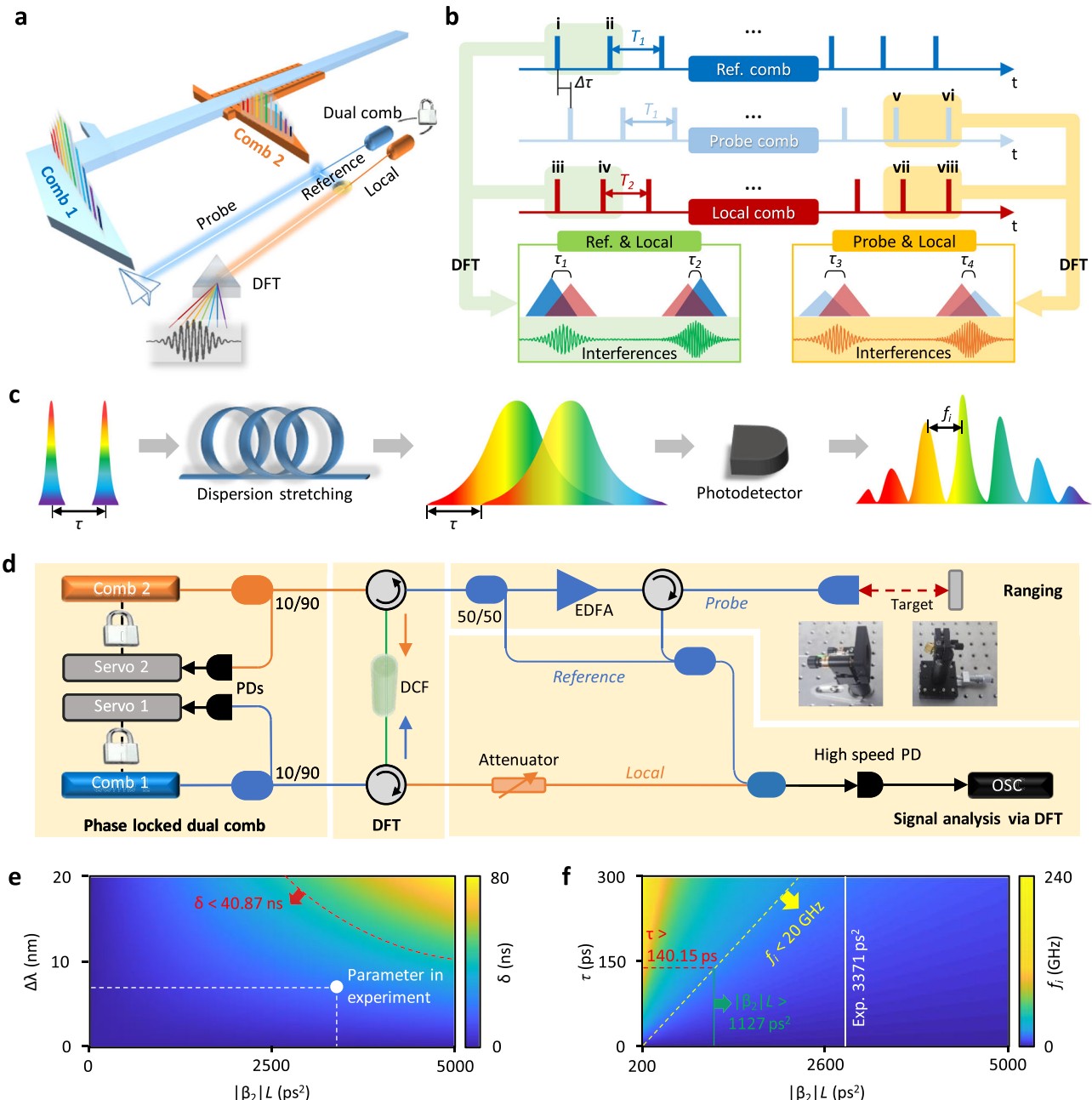

**Fig. 1 | Conceptual design and operation of the DFT based dual-comb ranging.**
**a** Schematic configuration. Two phase-locked frequency combs serve as the exchangeable signal light and local light, the dual-comb TOF is analyzed by using the DFT. **b** Mechanism of the high precision time delay analysis using DFT. Slight mismatches of the pulses are tested via spectral interference. **c** Operation of the pulse stretch and Fourier transform. After transformation, the pulse delay $\tau$ is converted into interference fringes with frequency difference $f_i$. **d** Experimental setup. PD photodetector, DCF dispersion compensated fiber, OSC oscilloscope, EDFA erbium doped fiber amplifier. **e**, **f** Parametric spaces. Both the stretching ratio and the $f_i$ are determined by the group-delay dispersion $|\beta_2|L$.

The comb 1 is divided into two paths, one works as the probe and the other serves as the reference. The probe light is used for ranging based on the TOF mechanism. The signal pulse (comb 1) and the local pulse (comb 2) are launched into the same dispersion-compensating fiber (DCF) section in opposite directions for stretching, and then we amplify the signal pulse and divide it as the probe and the reference. The DCF provides a $\beta_2 = 110.94$ ps$^2$/km. All the pulses are time-stretched in this process. By using 30.39 km DCF ($|\beta_2|L = 3371.47$ ps$^2$), our combs with a 3-dB spectral bandwidth $\approx 6.84$ nm (368 fs duration) can be stretched to 18 ns. Finally, the local-probe-reference interferogram is detected by a fast photodetector (25 GHz) and analyzed in a high-speed oscilloscope (20 GHz analog bandwidth). We further note that this approach extends the capabilities of precision measurement

from pulses with transformation limits to those with chirp, thereby providing a sweeping-free effect. This advancement proves beneficial in additional metrology systems, including those based on frequency-modulated continuous waves and optical frequency domain reflectometry.

Figure 1e, f discusses the calculated parametric spaces. In Fig. 1e, we show the $\delta$ linearly increases with $|\beta_2|L\Delta\lambda$, with a fixed $\lambda = 1550$ nm. In our DFT system, $\delta$ must be smaller than repetition period of the probe comb (40.87 ns), as the dashed red curve marks. Here the white dot presents the selected parameter in our experiment. In Fig. 1f, we map the relationship between the $|\beta_2|L$, the $\tau$, and the $f_i$. On one hand, $f_i$ is proportional to $\tau$, with a coefficient $|\beta_2|L$. In experiment, the maximum detectable $f_i$ is determined by the bandwidths of the

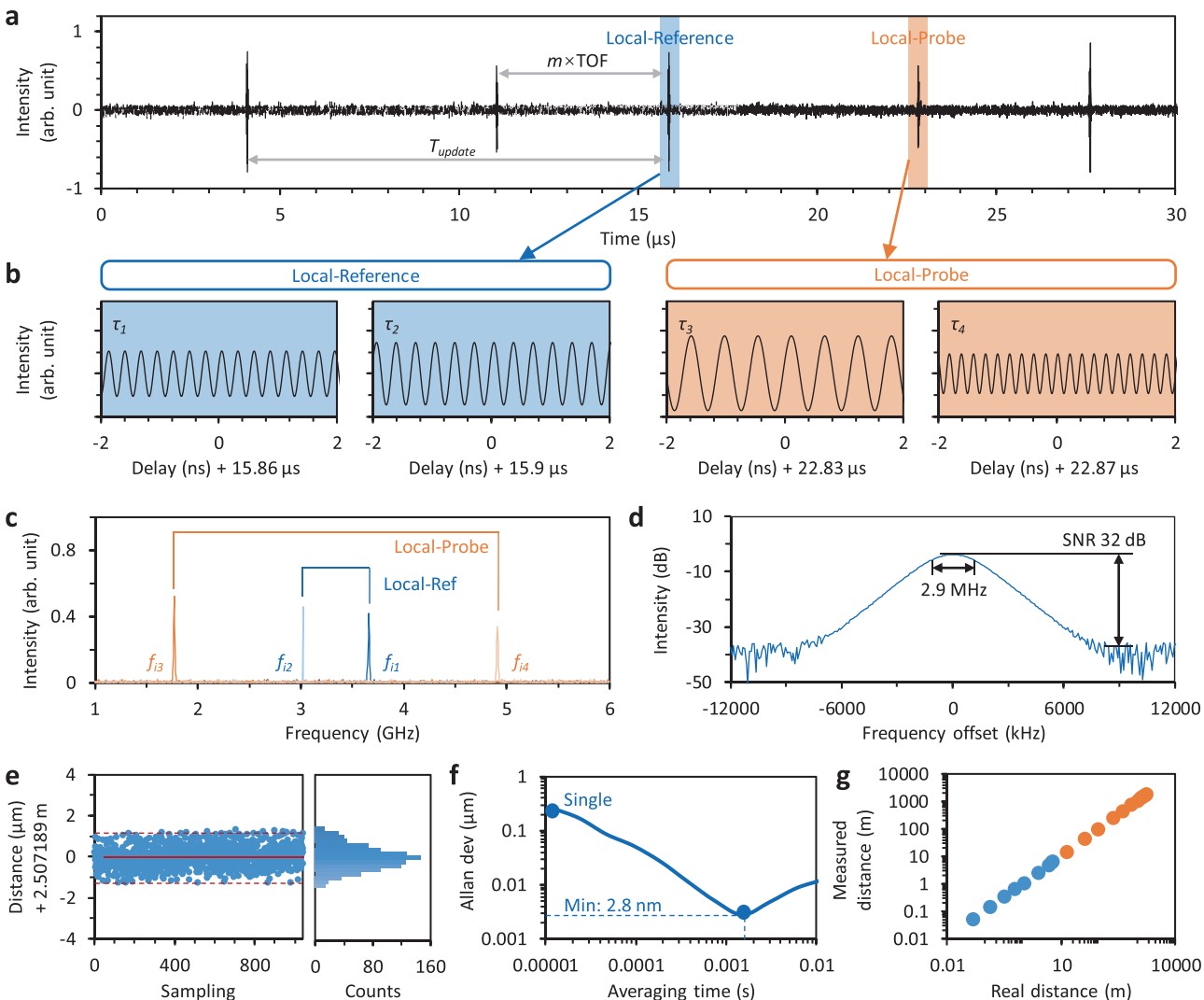

**Fig. 2 | Single point ranging. a** Measured interferogram of our dual-comb ranging system, the periodic pulse enhancements demonstrate the macroscopic information of distance, which relates to the $m \times$ TOF. **b** Zoom-in of the 'Local-Reference' peak and the "Local-Probe" peak. Due to the DFT, minor pulse-to-pulse delay-based fringes are clear. **c** Fast Fourier transform spectra of the fringes, here frequencies $f_{i1}$ to $f_{i4}$ correspond to the delays $\tau_1$ to $\tau_4$. **d** Zoomed-in spectrum of the $f_{iI}$ peak, with 3-dB bandwidth 2.9 MHz and SNR 32 dB. **e** Repeated ranging results and their distribution, suggesting a maximum error $\pm 1.3$ µm and an RMSE 288.7 nm. **f** Precision of the distance measurement versus averaging time. Allan's deviation reaches 2.8 nm with an averaging time 1.5 ms. **g** Correlation of the measured distance and the real distance. Here the blue dots show the range using one comb with 24.465 MHz, while the orange dots show the expansion using Vernier dual combs.

oscilloscope ($B_o = 20$ GHz), as the yellow dashed line marks. On the other hand, we must ensure that the detectable $\tau$ should be larger than the period difference $|T_1 - T_2|$(141.5 ps), the red dashed line shows this requirement. Hence, the minimum $|\beta_2|L$ is $\Delta f_{rep}/(2\pi f_{r1}f_{r2}B_o) = 1127$ ps², as the green solid line illustrates. In our experiment, we set $|\beta_2|L = 3371.47$ ps² (white solid line), located in the working region. More calculations are shown in Supplementary Note S1.3 (Fig. S2).

For a target with a fixed position, the interferometric trace of the synchronized pulse trains (including the returned probe, the reference, and the local pulses) is shown in Fig. 2a. In a macroscopic view, it appears similar to the results based on conventional dual-comb ranging method, one can roughly estimate the distance by $T_{update}$ and the enlarged TOF. Within the TOF duration, we see $N = 171$. But in a microscopic view, we can see more detailed information, as every pulse has been temporally stretched due to the large dispersion, and interferences occur in the pulse-to-pulse overlapping regions. When zooming the time in, we demonstrate the interferometric details in Fig. 2b. Due to the DFT, one can observe both the 'local-reference' fringes plotted in the blue boxes and the 'local-probe' fringes plotted in the orange boxes. These fringes reflect the slight pulse-to-pulse

mismatches ($\tau$). Specifically, the fringe frequencies corresponding to $\tau_1$ and $\tau_2$ are $f_{i,1} = 3.66$ GHz and $f_{i,2} = 3.02$ GHz; meanwhile, the fringe frequencies corresponding to $\tau_3$ and $\tau_4$ are $f_{i,3} = 1.77$ GHz and $f_{i,4} = 4.91$ GHz. Referring the Eq. (3), we can accurately obtain $\tau_1 = 77.52$ ps, $\tau_2 = 64$ ps, $\tau_3 = 37.48$ ps, and $\tau_4 = 104.04$ ps. These values meeting will with the period difference of the comb 1 and the comb 2 (141.52 ps). Therefore, referring Eq. (1), the measured distance is 2.507189 m. Figure 2c shows the fast Fourier transform spectra of these fringes ($f_{i,1}$ to $f_{i,4}$). We also note that in our measurement, $\tau_1 + \tau_2 = \tau_3 + \tau_4 = \Delta T$ is a fixed number. In extreme cases, the maximum $\tau$ may equal to $\Delta T$ (141.52 ps). Referring Eq. (3), the bandwidth of our photodetector & oscilloscope must be higher than 6.684 GHz. More in detail, we zoom one of the Fourier transformed peaks ($f_{i,1}$) in Fig. 2d for example. Thanks to the high stability of the interference, linewidth of each Fourier peak is <3 MHz, and typical signal-to-noise ratio (SNR) of each frequency component is > 30 dB. In ranging applications, such a performance suggests an in-principle resolution on 260 nm level in single shot measurement. In practice, the system performance is mainly limited by the stability of the dual-comb source. More discussion is shown in Supplementary Note S3.5. Moreover, the SNR of the

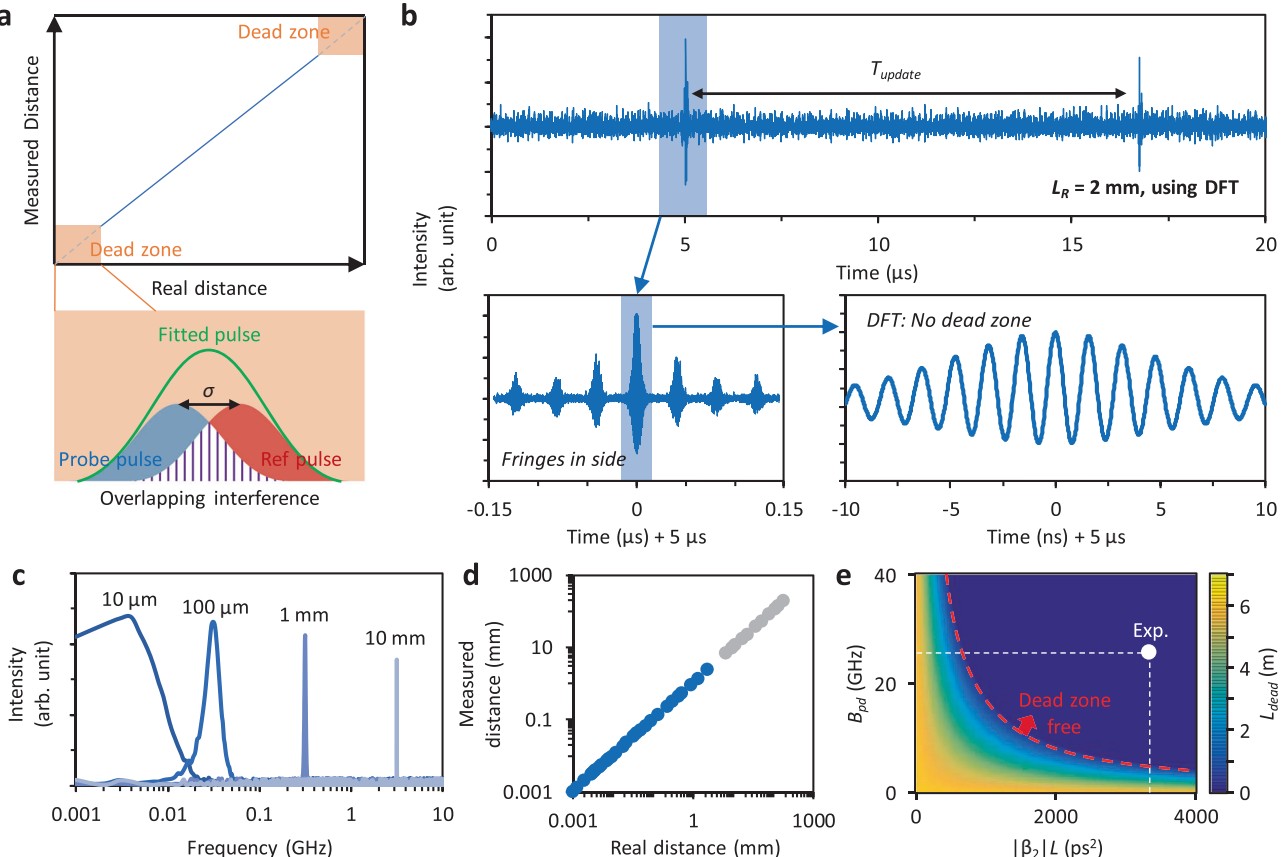

**Fig. 3 | Elimination of dead zones. a** Dead zones are regions where the probe and reference interferograms heavily overlap so that they cannot be distinguished from each other via pulse fitting, but such an overlap is measurable by using DFT. **b** Measured interferometric traces, here the to-target distance is 2 mm and the almost completely overlapped two pulses form clear interference fringes due to the DFT. **c** Fast Fourier transform spectra when the to-target distance increases from 10 μm to 10 mm. **d** Measurable region. The minimum detectable distance of DFT-based dual-comb ranging is only determined by the detection accuracy. **e** Parametric limitation for reaching the dead zone free measurement.

lines is commonly higher than 30 dB, suggesting that our measurement can effectively resist the intensity jitter of the pulses.

Then we record 1041 repeated ranging results in Fig. 2e. For a single-shot detection, the maximum error is ±1.3 μm, majorly limited by the instability of the fiber links. Here we also analyze the distribution of the data, which follows a Gaussian distribution. Standard root-mean-square error (RMSE) of these points is 272.5 nm. Figure 2f plots the statistical Allan deviation of our ranging measurements. For single-shot measurement, the typical detection limit is 262 nm. After 1.5 ms averages, the detection limit approaches 2.8 nm. Figure 2g shows the ranging distance. For a single comb with a repetition rate of 24.465 MHz, the NAR is about 6.1 m, which is limited by its pulse period. But after exchanging the role of dual-comb, the NAR extends to 1.75 km, determined by the dual-comb repetition rate difference $\Delta f_r$. These results show that the measured linearity and consistence of the dual-comb-based Vernier ranging system are good. In Supplementary Note S3.3 (Fig. S9), we compare our ranging system with the conventional TOF method, using the same dual-comb light source.

Moreover, the DFT-based dual-comb ranging can uniquely eliminate the "dead zone", which has been a widely recognized challenge in any ranging methods relying on TOF. For a typical TOF measurement, when the probe pulse and the reference pulse are close to each other (the distance approaching 0 or an integer multiple of the $L_{NAR}$), one cannot distinguish them due to the Rayleigh criterion. We show the dead zones schematically in Fig. 3a. As a consequence, pulse fitting won't make sense. To solve this problem, orthogonal polarization multiplexing was applied but introduced additional complexity[46]. In

addition, if the detected pulse has deformation (i.e., not in a perfect sech[2] or Gaussian shape), it will be more difficult to deterministically fit its peak position.

However, in our scheme, by analyzing the TOF via DFT-based interference, we smartly address this problem. Figure 3b plots a case in experiment, here we set the real distance $L_R$ = 2 mm, resulting in a temporal offset of the probe-reference pulses of 13.3 ps. Typically, the small offset between the "local-reference" peak and the 'local-probe' peak is difficult to retrieve. But our DFT scheme can distinguish the minor pulse-to-pulse offset clearly since we can measure the interferometric fringes in the broadened pulses. By zooming the above-measured traces in, we demonstrate that the result based on the DFT demonstrates clear interferometric fringes, illustrating a $f_i$ = 628.2 MHz. This verifies $L_R$ = 2 mm accurately. Specifically, Fig. 3c plots the fast Fourier transform spectra (in log scale) of more cases, where the frequencies of fringes increase when the ranging distance scales from 10 μm to 10 mm. Most of these cases are in the dead zone of a conventional TOF-based dual-comb ranging system. In our scheme, the minimum measurable distance is just determined by the minimum detectable $\tau$, which is <1 fs, as Fig. 2 displays. Therefore, the DFT-based dual-comb ranging is in-principle 'dead zone free'. Finally, we show the measured distance versus the real distance in Fig. 3d. The DFT-based dual-comb ranging enables linear and reliable measurements down to sub μm level.

In Fig. 3e, we explore the parametric boundaries necessary for eliminating dead zones. The unmeasurable area length, $L_{dead}$, in DFT-based demodulation is influenced by the comb repetition intervals $T_1$ and $T_2$, the photodetector's bandwidth $B_{pd}$, and the total dispersion $|\beta_2|$

$L$, as outlined in the equation:

$$L_{dead} = \frac{c}{2}\left[T_1 - \frac{4\pi T_1 |\beta_2| L B_{pd}}{T_1 - T_2}\right] \qquad (4)$$

It is evident that an increase in $|\beta_2|L$ results in a decrease in $L_{dead}$. Achieving $L_{dead}$ signifies that the system is free of dead zones, as red dashed curve shows. In our experiments, with $B_{pd} = 25$ GHz, $T_1 = 40.87$ ns, $T_2 = 40.73$ ns, the employment of a $|\beta|L = 3371.47$ ps² is significantly above the requirement for achieving a "dead zone free" state (white dot).

## Discussion

Based on technological optimization, our DFT-based dual-comb ranging system exhibits modular capability and long-term reliability, enabling application validation both inside and outside the laboratory (Supplementary Note Fig. S11, Fig. S12). First, taking advantage of the high accuracy and high consistence of the DFT-based dual-comb ranging, we show its capacity for micro-nano morphology. We have prepared a set of three-dimensionally carved English letters (UESTC), as Fig. 4a shows. Size of each letter block is about 2.2 cm × 2.2 cm. The thickness of the substrate is 3 mm. On the substrate, heights of the "U", "E", "S", "T", "C" are 0.1 mm, 0.2 mm, 0.3 mm, 0.4 mm, and 0.5 mm respectively. By using collimating lens, we scan our probe beam on the blocks, with spatial scanning step 1 mm. The 2D scanning operation is automatically controlled on an electric displacement table. Figure 4b shows the measured morphologic map (110 × 24 pixels), with an acquisition time of 0.02 ms for each pixel. In detail, we plot the measured height along the line 12 (marked by the red dashed line) and along raw 55 (marked by the yellow dashed line) in Fig. 4c. The measured heights precisely refer the real height of the letters. In Fig. 4d, we show the statistics of repeated measurements. Here we select five points, on the surface of "U", "E", "S", "T", and "C" respectively. The five points' coordinates are <7,12>; <30,5>; <55,12>; <78,12>; and <95,12>. The maximum measurement offset is ± 370 nm, and RMSE of the five selected points are 317 nm, 311 nm, 326 nm, 315 nm, and 331 nm, respectively.

The DFT-based dual comb ranging is also applicable for the dynamic measurement of high-speed moving objects, such as a turbofan rotor, as schematically shown in Fig. 4e. In the laboratory, we use this tool to monitor the fast rotation of fan blades for example. The electrical fan has 4 metal blades, its maximum rotation diameter is 10 cm, and the speed of the fan is ≈ 3000 rev/min. Each blade is tilted, with a tilt height ≈ 1 cm. On each blade, there is a small protrusion (≈530 μm), marked by the red dashed curve. We focus the laser on the blades at diverse locations ($R = 0$ mm and 40 mm). For enhancing optical sensitivity, the probe comb is amplified by using a low-noise EDFA and collimated by using a lens. Figure 4f plots the measured distance when the probe comb continuously irradiates the fan blades. Determined by the dual-comb repetition rate difference, sampling rate is 85 kHz. Within 20 ms, the motion trajectories of the 4 blades are captured. At $R = 40$ mm, we measure each blade shows a height difference 9.6 mm during rotation. Here the tiny protrusions on the fan surface are also well detectable. In the dynamic measurement, for every rotation period, root mean square for blades 1–4 reaches 68 μm, 62 μm, 73 μm, and 71 μm respectively. The Allan deviation suggests that when the averaging time is 1.2 ms, the measurement accuracy approaches 170 nm (Fig. 4g). We note that the measurement accuracy of the dynamic ranging is lower than the accuracy when detecting a static target since the moving blades itself is not so stable.

In addition to in-lab measurements, we also validate the effectiveness of our DFT-based dual comb LIDAR for field applications. For example, this method provides a tool to detect small and slow-moving targets, such as unmanned aerial vehicles (UAVs), which are important targets in both military and civilian applications but are often challenging to measure with traditional radar techniques. In Fig. 5a, b, we demonstrate the experimental scenario. We employ a collimator on a tripod head (HY-LW18-01A, horizontal rotation $\alpha = 0$–360°, pitch rotation $\beta = 0$–90°, accuracy 0.1°) to transmit and receive pulses from the probe comb and affix a cube-corner prism on the UAV (DJI Air 3) to enhance light reflection. Alternatively, one can increase the output power of the probe comb via amplification or use a receiver with a larger numerical aperture. Figure 5c illustrates the measured relative position of a quasi-static target, specifically when the UAV is hovering. In this case, spatial location of the UAV is $\alpha = -15°$, $\beta = 45°$, and the LIDAR to target distance is 39.32 m. In Fig. 5d, we plot the continuously collected distance data. It demonstrates that the hovering UAV is rapidly shaking (ms scale, mainly due to the mechanical vibrations) while slowly drifting (second scale, mainly determined by the wind force). Besides measuring a quasi-static target, by scanning the tripod head, we can also trace a dynamically moving target. For instance, in Fig. 5e, we control the UAV linearly flying towards our LIDAR, from the location P1 (coordinate: $x = 11.3$ m, $y = 25.2$ m, $z = 19.4$ m) to P2 (coordinate: $x = 7.1$ m, $y = -4.4$ m, $z = 2.6$ m). In this process, the LIDAR-to-target distance decreases from 33.751 m to 8.748 m. Figure 5f shows the measured results. From frame 0 to frame 5, we record the polar coordinate information of the UAV during flight, as Fig. 5g illustrates.

In conclusion, by introducing the dispersive Fourier transform to dual-comb-based LIDAR, we demonstrate a physical paradigm that coherent laser ranging can merge the benefits of time-of-flight and in-pulse interferometric methodologies.

Different from conventional methods reliant on temporal pulse separation retrieval, the DFT-based dual-comb LIDAR precisely examines Fraunhofer's diffraction in spectrum, thus provides a solution to overcome the compromise between speed and accuracy in dual-comb ranging. This creates a powerful tool for determining absolute distances, boasting high accuracy down to 2.8 nm, extensive NARs up to 1.7 km, and a unique capability for dead-zone-free measurement. Utilizing this DFT-empowered dual-comb ranging scheme, we demonstrate its promising potential in undertaking high-precision morphological measurements and dynamic motion detection. By combining strengths of the Vernier effect of dual comb, time-of-flight techniques, and interferometric analysis, this DFT-based dual-comb ranging approach could pave the way for devising advanced LIDAR systems with broader applications, such as positioning, imaging, and deformation analysis.

## Methods

### Mechanism of the in-fiber dispersive Fourier transform

The large GVD is provided by two DCF modules (Corning PureForm DCM-D-080-04) with a dispersion parameter of $D = 87$ ps/nm·km and a dispersion slope of $S = 0.025$ ps/nm²·km. Each module has a group delay dispersion (GDD) of −1322 ps/nm with a total loss of 6.86 dB. Fiber length of each module is 15.2 km. The two DCF modules are positioned within a drying chamber that maintains a constant temperature and humidity environment. This chamber is situated on an active vibration isolation platform to ensure stability. To suppress the ultrafast pulses' cross-phase modulation (XPM) in the DCFs, we pump signal combs and reference combs from opposite directions through two circulators. Then, two low-noise erbium-doped fiber amplifiers (EDFA) with a gain up to -45 dB are additionally implemented to provide sufficient signals to detection. After time stretching, the pulse width of a femtosecond pulse (368 fs) with a bandwidth of 6.84 nm is extended to 18 ns. Here, we choose a pulse repetition frequency ≈ 24.55 MHz to ensure that the adjacent pulses do not temporally overlap each other. We note that in addition to DCF, other components can also be used to provide the dispersion. For instance, customized chirped fiber Bragg gratings (CFBGs), capable of offering substantial dispersion over short distances, can be utilized to significantly enhance the system's integration.

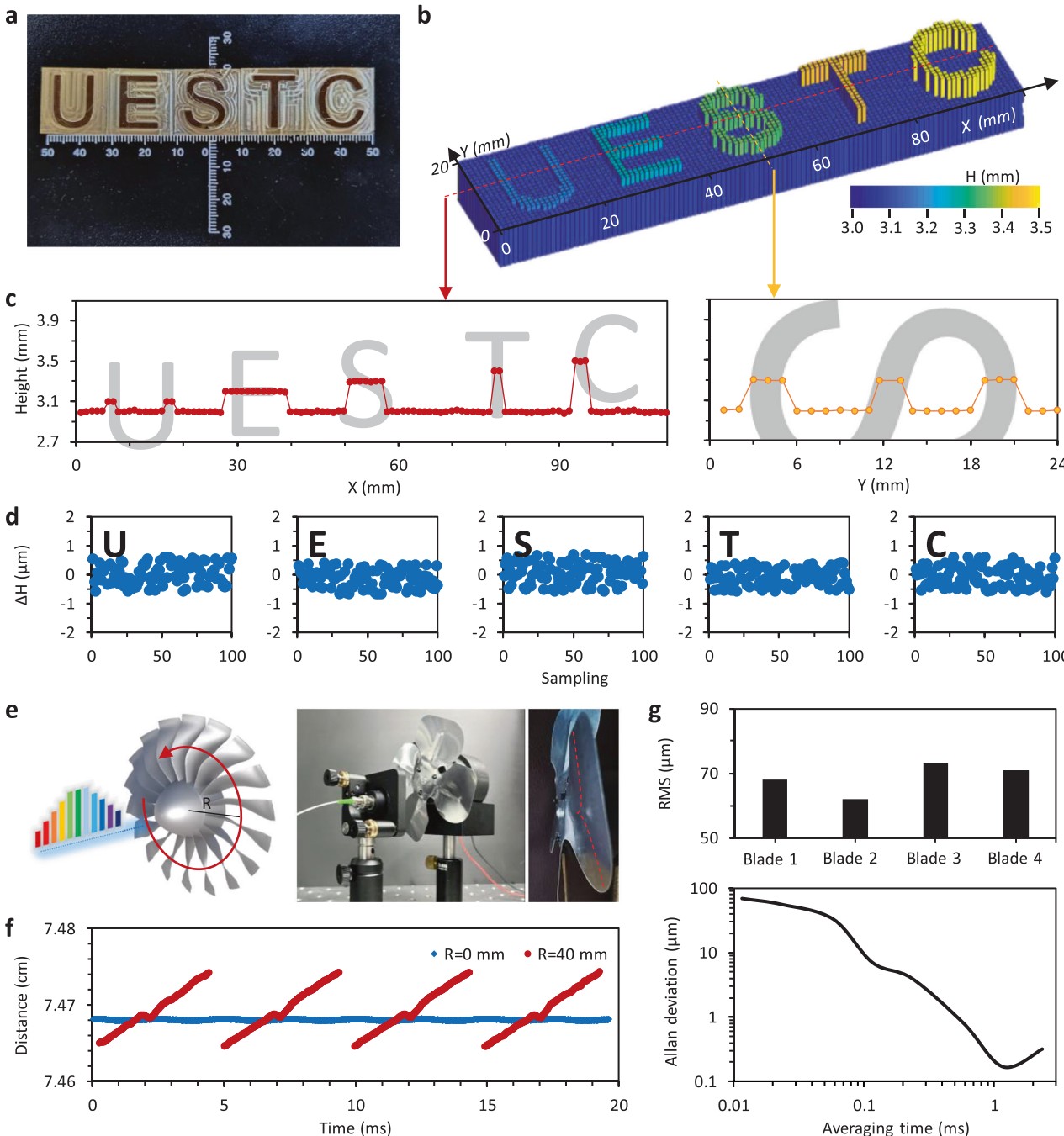

**Fig. 4 | Static morphology and dynamic displacement detection using the DFT-based dual-comb ranging. a** Picture of the to-be-tested sample. **b** Measured morphologic map (110 × 24 pixels) of the sample. Here the colored columns show the height at different points. **c** Two measured profile curves, showing the heights in the *x* direction and the *y* direction. **d** Repeated measurements. **e** Conceptual design and picture show the DFT-based dual comb tool measures a rotating fan. **f** Measured displacements of a rotating fan, the blue dots show the central location (*R* = 0 mm) while the red dots show the blades (*R* = 40 mm). **g** Performance of the dynamic ranging. Maximum RMS is 73 μm, and the typical Allan's deviation suggests a detection limit 170 nm at averaging time 1.2 ms.

## Mode-locked fiber laser-based dual soliton frequency combs

We use dual mode-locked fiber lasers as the signal comb and the reference comb respectively. The fiber resonators are based on hybrid passive mode-locking mechanism (NPR + SESAM). Each mode-locked laser comb contains a 980 nm pump laser diode, a section of ~2 m long erbium-doped fiber (EDF), a semiconductor saturable absorber mirror (SESAM), a section of ~6 m long single-mode fiber, a polarization controller (PC) and a polarization dependent-optical integrated component (PD-OIC). The intracavity power is separated by an 80/20 fiber beam splitter, here the 20% power outputs. By adjusting the polarization in cavity, we obtain

mode-locked pulse sequence with an average output power of 0.5 mW when the pump power is 10 mW. The total cavity lengths of the signal comb and the reference comb are 8.175 m and 8.147 m respectively, corresponding to the repetition rate of signal pulses and reference pulses 24.465 MHz and 24.55 MHz. Temporal pulses of the dual combs have a single pulse width ~368 fs / 416 fs (local comb/ signal comb). The time-bandwidth product (TBP) is 0.316 for local comb and 0.315 for signal comb, which is close to the transformation limit, verifying that the initial pulses are chirp-free. We control the geometric length *L* of the fiber cavity by using a piezoelectric transducer (PZT) with a maximum diameter expansion 0.38 μm

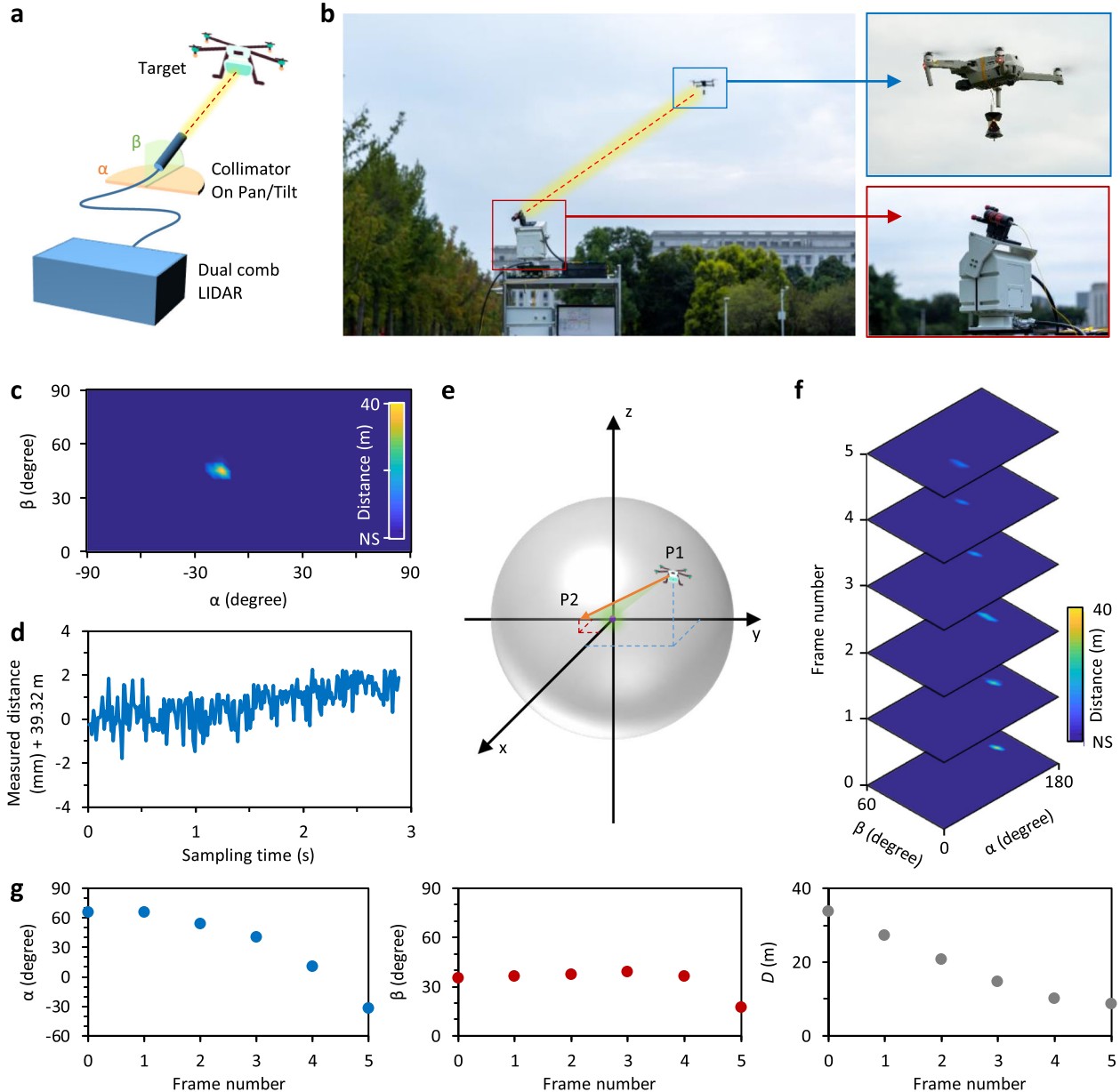

**Fig. 5 | Detection of a UAV out of lab. a** Concept and experimental schematics. **b** Pictures of the field test. The probe comb pulses are transmitted and received by a collimator fixed on a tripod head, the target is a typical 4-axis UAV. **c** position information of a hovering UAV, which is regarded as a quasi-static target. Color bar: distance. **d** Distance between the dual-comb LIDAR and the target in continuous measurement. **e** The scenario that the LIDAR traces the UAV moving from P1 to P2 spatially. **f** Measured frames during the UAV movement. Color bar: distance. **g** During the flight of the UAV, we record its spatial parameters $\alpha$, $\beta$ and distance $L$.

when the driving voltage is 150 V. By using two servo controllers (Vescent D2-125-IP-230), we locked the repetition rates to a stable RF source.

### Experimental setup and device details

Before transmitting the probe pulse into free space, we used a low noise EDFA to boost the comb power (up to 2 W in average, using the chirped-pulse-amplification technique). The collected diffuse reflection ratio is about −26.5 dB. Before the signal pulses, reference pulses, and local pulses are received by the detector, they pass through a PC respectively to ensure that the modulation depth of the interference fringes is large enough. The interference signal is detected with an AC-coupled 25-GHz InGaAs photodetector (Model 1414, Newport) and digitized with a high-speed 20-GHz oscilloscope (DPO72004B, Tektronix) at a sampling rate of 50 GS/s.

## Data availability

All data are available in the main text or the supplementary materials.

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

## Acknowledgements

The authors acknowledge support from the National Key Research and Development Program of China (Grant 2021YFB2800602), the National Natural Science Foundation of China (Grant U2130106 & 62305050), the National Postdoctoral Innovation Talent Support Program of China (Grant BX20220056) and Guangdong Basic and Applied Basic Research Foundation (Grant 2024A1515011665). We acknowledge Zeping Wang for his technical support during the experiment.

## Author contributions

B.C., T.T., and J.D. contributed equally to this work. Y.R. and B.Y. led the group. B.Y. and B.L. led this study. B.C., T.T., J.D., and Z.L. contributed the experimental investigations. B.C., C.W., H.X., and Z.W. built the laser comb devices. B.C., T.T., X.H., and Y.L. optimized the experimental setup. B.C., T.T., J.W., and T.Z. contributed dynamic motion detection. B.C., B.Y., T.T., K.K.Y.W., L.K., and B.L. contributed the theoretical analysis. All authors processed and analyzed the results. B.C., B.Y., T.T., B.L., and Y.R. prepared the manuscript.

## Competing interests

The authors declare no competing interests.
