## [Peer Review File · Nature Communications]

REVIEWER COMMENTS

Reviewer #1 (Remarks to the Author):

The authors report a dispersive Fourier transform (DFT) based LIDAR method utilizing dual soliton laser combs enabling absolute distance measurements with precision starting from 262 nm in single shot to 2.8 nm after averaging 1041 times. The topic of research is interesting and timeliness.

However, I believe the novelty of the results presented in this manuscript is not enough for Nature Communications.

1) The absolute distance measurements using dual comb has been published in Nature Photonics in 2009 [ref. 29] (<https://www.nature.com/articles/nphoton.2009.94>).

The claimed advantage of this work is the high precision of 262 nm which also achieved by Dingtong Hu et. al. (<https://www.sciencedirect.com/science/article/abs/pii/S0030401820309846#:~:text=A single free%2Drunning mode%2Dlocked Er%2Dfiber laser,and mixed with local comb>).

Obviously, the DFT has been used in single shot measurements by K. Goda and B. Jalali (<https://www.nature.com/articles/nphoton.2012.359>) and Julien et. al (<https://www.nature.com/articles/nphoton.2008.293>).

2) Manuscript has a lot of technical details without appropriate outline of the suggested technology to make it clear to wider community. The claimed (see point 1 above) advances vs traditional dual comb distance ranging, e.g., “with precision starting from 262 nm in single shot, to 2.8 nm after averaging 1041 times (1.5 ms), in a non-ambiguity range over 1.7 km” are provided by using an expensive module to support DFT. Unlike the DFT-based distance ranging, simpler dual-comb technique (Coddington, I., Swann, W., Nenadovic, L. et al. Rapid and precise absolute distance measurements at long range. Nature Photon 3, 351–356 (2009). <https://doi.org/10.1038/nphoton.2009.94>) can support a highly competitive specs:

“...we demonstrate a coherent laser ranging system that combines the advantages of time-of-flight and interferometric approaches to provide absolute distance measurements, simultaneously from multiple reflectors, and at low power. Through the optical carrier phase, the precision is improved to better than 5 nm at 60 ms, and through the radio-frequency phase the ambiguity range is extended to 30 km, potentially providing 2 parts in 10¹³ ranging at long distances.”

3) Specific point on the text: Abstract: P.1

“DFT-based spectrally interferometric measurement instead of pulse-fitting in time domain...”

The phrase is incorrect: As follows from the reference mentioned above, dual-comb distance ranging is using TOF and interferometric technique based on processing data in a Fourier domain.

Overall, this manuscript lacks novelty to meet criteria of Nature Communications. I recommend to reject this paper.

Reviewer #2 (Remarks to the Author):

In this article, the authors innovatively employed dispersive Fourier transform (DFT) in a dual-comb ranging system and demonstrated its application potentials in and out of lab setting. The combination of advanced in-pulse interferometric methodology and dual comb metrology disrupts the Nyquist sampling constraint while nullifies the trade-offs between speed and precision in dead-zone-free ranging. Experimental results indicated impressive performances and the manuscript is generally well organized. I strongly recommend this work to publish in Nature Communications, after the following questions were well addressed.

1. The authors used fiber laser combs. In reference to integrated microcomb based ranging systems (e.g. in Chang, L. et al. Nat. Photon. 16, 95 2022; Shu, H. et al. Nature 605, 457, 2022), does the fiber combs show unique advantages in principle or technique?
2. Comb ranging methods are known for their versatility. Is the DFT applicable only in TOF? The authors should discuss whether this scheme could be implemented in other comb ranging systems, such as OFDR (e.g. in Chen, R. et al. Nature Photon. 17, 306, 2023).
3. In the maintext, the authors mainly focus on articulating their own results. To benefit a wider audience not familiar with this field, it may be helpful to include a comprehensive table that contrasts the primary indices with existing literature.
4. This scheme necessitates a large dispersion element (i.e., a DCF with a length of 15.2 km), which could limit its integration for broader applications. Could the authors discuss potential optimizations? Alternatively, could the authors suggest any substitutes (e.g., crystals) that might achieve a similar effect in the future?
5. 1. In relation to the previous question, how can timing jitter in the long fiber be mitigated or suppressed in an open environment? The authors should provide experimentally measured data.
6. To stabilize the dual comb, the authors used two servos, this increases the complicity in system. Referring other laser stabilization schemes (e.g. J. Bowers, et al. eLight 3, 1, 2023), can the authors discuss a way to simplify their feedback loops?
7. In dual comb ranging, reflected optical power plays a pivotal role in ensuring the Signal-to-Noise Ratio (SNR). It would be beneficial if the authors could measure this reflected power from varying distances and discuss whether an optical amplifier is necessary or not.
8. The application of drone detection outside the laboratory mentioned by the authors is quite intriguing. However, I am curious about the success rate of signal reception in this context. Additionally, while the use of a corner prism to secure light reflection is appreciated, it is important to note that in real-world applications, a corner prism may not be present on every target. Therefore, the authors need to elaborate on potential solutions to this hurdle.
9. The textual elements in some figures within the article seem inconsistent, and there are typographical errors scattered throughout. I would suggest that the authors diligently refine their manuscript before submission for acceptance.

In sum, this research presents an innovative physical mechanism for rapid and precise LIDAR signal demodulation. The authors offer substantial results that validate the applicability of this method, both within and outside the lab confines. Given its timeliness and broad appeal, I would recommend publication in Nature Communications, contingent upon the authors' willingness to make major revisions to their manuscript.

Reviewer #3 (Remarks to the Author):

The paper “Dispersive Fourier transform based dual-comb ranging”, by Chang et al., reports on the implementation of a dual frequency comb system for optical ranging. Specifically, the authors leverage fibered dispersive Fourier transform for time-stretching the frequency combs. Instead of using time-of-flight measurement approaches for optical ranging, the proposed system takes advantage of an interferometric technique to retrieve absolute distance measurements.

In this paper, the implemented system takes advantage of a particular approach combining asynchronous dual comb measurements based on two soliton phase locked lasers to provide the required Vernier effect, along with dispersive Fourier transform to achieve temporal encoding of the comb spectra in the temporal domain. The latter allows for obtaining a time-dependent interferometry between the different combs (local-reference and local-probe) that are used to achieve efficient distance measurements with nanoscale resolution. The proposed technique allows for extended non-ambiguity range and drastically reduced dead-zone that are common issues in most LIDAR ranging techniques. The results are convincing and proof-of-concept demonstrations are illustrated via two practical examples (for out of the lab ranging, and high-resolution dynamic morphology measurements)

Overall, the paper is scientifically sound and innovative enough to be considered for publication in Nature Communications. I think that the proof-of-concept demonstrations are significant, and the overall approach is promising. I however believe that several points, appended below, should be addressed before considering the manuscript for publication.

1/ I appreciated the overall scientific quality of the paper but the writing could be reasonably improved by thorough proofreading.

1a/ First, several sentences are maybe overselling/general and do not bring much while being present at the expense of the overall clarity of the sentence. I would recommend rephrasing these for the understanding of the reader. Some examples:

- Line 43: Recently, the comb stabilization technology has pushed the light towards the level of “photonic clock”...
- Line 249: the DFT-based dual comb LIDAR scrutinizes Fraunhofer's diffraction with exceptional resolution, shattering the Nyquist sampling constraint while nullifying the trade-off between speed and precision in ranging.

1b/ Some wordings also appear poorly chosen and requires the reader to look into/guess what was meant. This should be addressed for the scientific clarity of the paper. Some examples:

- Line 43: integral ?
- Line 54: suppresses => suppressing
- Line 74: transforms the time-of-flight based ranging mechanism from temporal separation to spectral interferogram, and thus overcomes the trade-off between speed and accuracy (understandable, but unclear
- Line 76: Analyzing
- Line 84: swapping => sweeping ?

- Line 98: T1 and T2 are not really defined in the main text
- Line 190: In our scheme, the minimum detectable τ is mainly determined by the frequency resolution of the Fourier transform peak (hundreds of nanometers) => What does hundreds of nanometers refer to?

2/ Figure 1 b is heavy and can be unclear to the reader (what does τ_i refer to ?). I think integrating the local reference overlap in blue and explaining things a bit better may provide further clarity to the reader (how is the time offset measured). Maybe integrating Fig. S1 in the main text could be useful?

3/ The real scale of Fig 2c is missing and the zoom does not allow identifying the beatnote bandwidths. It is therefore hard to understand where/how the 3 MHz bandwidth and below 90 kHz resolution are obtained. Can we get an idea of the SNR of the beatnote vs the noise floor or the potential presence of other frequency components in the RF spectrum ?

4/ In the supplementary, I believe that the authors may explain further how the ambiguity range is extended by “exchanging” the role of the combs as illustrated in Fig S9.

5/ Fig.1d-e needs additional information to be properly assessed and can be completed based on Fig S2.

6/ More generally, the supplementary document contains essential information regarding the dead zone management based on experimental parameters (chromatic dispersion, PD bandwidth, repetition rate). Therefore, some information contained on S1.3 should be recalled in the main document, for the reader to properly estimate the related limitations.

7/ Can the authors complete the experimental set-up description in the main document regarding the device compactness and long term stability, two criteria of main importance for embedded systems.

8/ Can the authors comment and discuss the sensitivity of this DFT-based ranging method? As the femtosecond pulse is temporally stretched, the power temporal (DFT spectral) density is fairly weak. As the diffuse reflection power is -17 dBm, and the DFT temporal bandwidth is of the same order of magnitude as the comb period, the peak power is almost equal to the average power and one would thus expect a visibility measurement based on $\sim 100 \mu\text{W}$ probe peak power detection (at best). What is the impact on the sensitivity in this case? What is the sensitivity threshold for measurable fringe visibility with sufficient SNR for suitable ranging? Any idea of the maximal achievable ranging length considering this decreased peak power considering absorption and beam divergence?

9/ While probably out of the scope of this study, could the author comment on the reliability/robustness of these measurements in non-ideal conditions? Here, the authors use reflective components for ranging (no stringent sensitivity issues) and base their analysis on the DFT interferometry considering ideal time-stretch. Can a lower reflectivity could be considered a suitable countermeasure for this implementation compared to pulsed TOF approaches? Can external factors (e.g. propagation in a dispersive medium like water etc.) or man-made signal disruption (signal modulation, etc.) have a stronger impact on this DFT dual-comb ranging technique rather than other approaches?

Responses to reviewers

Outline:

#1 Response to Reviewer 1	P01~P08
#2 Response to Reviewer 2	P09~P17
#3 Response to Reviewer 3	P18~P27

Colors:

Reviewers' comments (Black), Response (Blue), Action taken (Red)

Reviewer #1

The authors report a dispersive Fourier transform (DFT) based LIDAR method utilizing dual soliton laser combs enabling absolute distance measurements with precision starting from 262 nm in single shot to 2.8 nm after averaging 1041 times. The topic of research is interesting and timeliness. However, I believe the novelty of the results presented in this manuscript is not enough for Nature Communications.

General response:

Thank you for reviewing our manuscript. We are pleased to see that you find the topic of our research interesting and timely. In the revised manuscript, we have further emphasized the originality of our work and have sharpened our discussion to highlight the advantages our system offers compared to previous studies, while being careful not to overstate its performance. Below, we provide point-by-point responses to address your concerns.

1. The absolute distance measurements using dual comb has been published in Nature Photonics in 2009 [Nature Photon. 3, 351 (2009)]. The claimed advantage of this work is the high precision of 262 nm which also achieved by Dingtong Hu et. al. [Opt. Commun. 482, 126566 (2021)]. Obviously, the DFT has been used in single shot measurements by K. Goda and B. Jalali [Nature Photon. 7, 102 (2013)] and Julien et. al [Nature Photon. 3, 99 (2009)].

Response:

Thank you for this note. We fully agree that scientists have reported impressive ranging performance when using dual comb technology. Specifically:

- In the research article [*Nature Photon. 3, 351 (2009)*], Prof. Newbury's group raised the concept of dual comb ranging and achieved a precision of 5 nm at a 60 ms update rate. However, this

performance is achieved by using an expensive hydrogen maser as the RF reference. We give more discussion in the response of your question 2.

We fully recognize that this work is a pioneer in dual comb ranging. It is exactly because of this work that the trade-off between accuracy and speed in ranging technology has become an important scientific problem, while our work finds a new way to overcome this challenge.

- In the research article [*Opt. Commun.* 482, 126566 (2021)], Prof. Hu's group used a creative algorithm (Kalman filtering), and obtained a ranging precision of 225.7 nm, with a fixed update rate of 20 Hz.

However, it should be noted that the measurement precision is only 10 μm without averaging (~ 40 times worse than our work), with a measurement frame rate of 2 kHz (40 times slower than our work). Admittedly, using this algorithm to improve ranging accuracy is an effective strategy, but it also introduces additional system complexity and delay. Moreover, the Kalman filtering is equally adaptive to our system, which will further improve the ranging precision.

- In the review article [*Nature Photon.* 7, 102 (2013)], Prof. Goda and Prof. Jalali introduced that the DFT technology can uniquely achieve real-time measurement of fast non-repetitive events, and summarized its applications such as spectroscopy and biochemical detection.

We fully respect that Prof. Goda and Prof. Jalali are pioneers of DFT. However, this is the first time DFT and dual-comb metrology has been innovatively combined, resulting in a new LiDAR modality that provides high precision, high speed and dead-zone free performances simultaneously.

- In the research article [*Nature Photon.* 3, 99 (2009)], Prof. Picque's group leveraged the DFT and demonstrated a high resolution Fourier transform spectrometer.

This investigation fully demonstrated the high-precision sampling of frequency information using DFT technology. But it still has little to do with ranging technology.

In a nutshell, dual comb ranging and DFT technology were independently investigated, while our work, for the first time, find an interdisciplinary combination of them. Thanks to this combination, we demonstrate novel physical mechanisms:

(1) We show that minor pulse-to-pulse delay in dual-comb ranging can be demodulated in frequency domain rather than in temporal domain. This provides a new physical dimension for the analysis of ranging signals and enables extremely high precision.

(2) In extension, our method provides a deeper understanding for both dual-comb ranging with DFT. Conventionally, dual comb ranging was mainly realized by using transform-limited femtosecond pulses. In this work, we demonstrate that high precision ranging can be achieved by using time-stretched pulses with linear chirping. This innovation represents a paradigm extension and helps to relieve the demand for light sources in ranging technology.

Fig. R1 shows the difference in principle, between our method and a conventional dual-comb ranging method.

Fig. R1. Differences between our scheme and conventional dual-comb ranging.

We reinforce our **Fig. 1** in the maintext with new statements.

Fig. 1. Conceptual design and operation of the DFT based dual comb ranging. a, Schematic

configuration. Two phase locked frequency combs serve as the exchangeable signal light and local light, the dual-comb TOF is analyzed by using the DFT. **b**, Mechanism of the high precision time delay analysis using DFT. Slight mismatches of the pulses are tested via spectral interference. **c**, Operation of the pulse stretch and Fourier transform. After transformation, the pulse delay τ is converted into interference fringes with frequency difference f_i . **d**, Experimental setup. PD: photodetector, DCF: dispersion compensated fiber, OSC: oscilloscope. EDFA: erbium doped fiber amplifier. **e & f**, Parametric spaces. Both the stretching ratio and the f_i are determined by the group-delay dispersion $|\beta_2|L$.

2. The manuscript has a lot of technical details without appropriate outline of the suggested technology to make it clear to wider community. The claimed (see point 1 above) advances vs traditional dual comb distance ranging, e.g., “with precision starting from 262 nm in single shot, to 2.8 nm after averaging 1041 times (1.5 ms), in a non-ambiguity range over 1.7 km” are provided by using an expensive module to support DFT. Unlike the DFT-based distance ranging, simpler dual-comb technique [Nature Photon 3, 351 (2009)] can support a highly competitive specs: “...we demonstrate a coherent laser ranging system that combines the advantages of time-of-flight and interferometric approaches to provide absolute distance measurements, simultaneously from multiple reflectors, and at low power. Through the optical carrier phase, the precision is improved to better than 5 nm at 60 ms, and through the radio-frequency phase the ambiguity range is extended to 30 km, potentially providing 2 parts in 10^{13} ranging at long distances.”

Response:

Thanks for the suggestion that we need to make a clear outline for wider community. We fully agree that we should not oversell the detailed performances of our dual comb LIDAR, while we should highlight the unique features of our technique.

Related to the dual comb ranging scheme published in *Nature Photonics*, we summarize our unique technical advances here:

- **Lower cost:** We note that the technique reported in *Nature Photonics* (3, 351 2009) yields a precision of 3 μm with an ambiguity range of 1.5 m and speed 50 kHz, before ultra-stable optimization. Please see **Fig. R2**.

The ability to determine absolute distance to an object is one of the most basic measurements of remote sensing. High-precision ranging has important applications in both large-scale manufacturing and in future tight formation-flying satellite missions, where rapid and precise measurements of absolute distance are critical for maintaining the relative pointing and position of the individual satellites. Using two coherent broadband fibre-laser frequency comb sources, we demonstrate a coherent laser ranging system that combines the advantages of time-of-flight and interferometric approaches to provide absolute distance measurements, simultaneously from multiple reflectors, and at low power. The pulse time-of-flight yields a precision of 3 μm with an ambiguity range of 1.5 m in 200 μs . Through the optical carrier phase, the precision is improved to better than 5 nm at 60 ms, and through the radio-frequency phase the ambiguity range is extended to 30 km, potentially providing 2 parts in 10^{13} ranging at long distances.

Fig. R2. Screenshot of the abstract, Nature Photonics 3, 351 (2009).

To achieve the performance that “the precision is improved to better than 5 nm at 60 ms, and through the radio-frequency phase the ambiguity range is extended to 30 km”, the authors used an expensive hydrogen maser as the RF reference and took advantage of advanced phase interferometry algorithm. A hydrogen maser commonly costs over 500,000 USD. Please see **Fig. R3**. In comparison, we are confident that our ranging system is simpler and more cost effective.

The limit to the fractional accuracy in the time-of-flight and interferometric range measurements is ultimately the fractional accuracy in the rf timebase and optical frequency, respectively. Here, we rely on an rf time base (hydrogen maser) that can support better than 1×10^{-13} fractional ranging resolution, that is, 3 nm in 30 km or below the systematic uncertainty. The fractional accuracy of the carrier frequency will depend on the underlying c.w. reference laser, which can be stabilized to a calibrated

Fig. R3. Screenshot of the maintext (Page 354), Nature Photonics 3, 351 (2009).

- **Faster speed.** Breaking the Nyquist sampling constraint in traditional dual-comb system, our DFT based dual comb ranging method can achieve faster update rate (U_s). Specifically, U_s of a conventional dual-comb ranging must be smaller than $f_{r1}f_{r2}/2B$, here f_{r1}, f_{r2} are repetition rates of the two combs, B is the 3-dB spectral bandwidth of the combs. This is the premise of TOF based dual comb ranging. Afterwards, in dual comb ranging, the measured time (Δt) is much higher than the real TOF ($\Delta\tau$): $\Delta t = f_{r1}/|f_{r1}-f_{r2}|\Delta\tau$, here $f_{r1}/|f_{r1}-f_{r2}|$ is the stretching factor. For achieving a higher accuracy, one expects a higher $f_{r1}/|f_{r1}-f_{r2}|$, this for sure leads to lower speed. So, there is a trade-off between accuracy and speed. However, in our strategy, the measured TOF meets $\Delta\tau = T_2 - N(T_1-T_2) + \tau_2 - \tau_4$. The accuracy depends on the precision of $\tau_2 - \tau_4$, rather than the stretching factor. Therefore, we offer a way to break the trade-off between speed and accuracy for ranging.
- **Higher sensitivity:** Leveraging the pulse stretching strategy, we can efficiently pre-amplify the signal comb. This also introduces chirped-pulse-amplification (CPA) into ranging technique, enables a much higher output power, benefiting to higher sensitivity, higher SNR and larger ranging range.
- **Dead-zone free:** There is no dead zone in our ranging system. Generally, in a dual-comb ranging system based on TOF method, when the measured distance is close to the multiple integers of half of the distance L_{pp} corresponding to the pulse period, the interference signals of the reference and the probe will overlap in the time domain, resulting in non-measurement. The distance corresponding to the pulse width of the interference signal is called the ranging "dead zone". For instance, the work published in [*Nature Photon* 3, 351 (2009)] suffers the dead zone. In our system,

due to the different mechanism, even if the two pulses almost coincide, we can still obtain the time delay information precisely through the interference fringe.

Another concern from the reviewer is whether we used an expensive module to support DFT. In our proposed DFT-based dual-comb ranging system, most of the cost is from the need for broadband photodetectors and high-speed acquisition cards. Here, the bandwidth of both devices is determined by the maximum frequency of the stripes. As shown in **Fig. 1e** in maintext, the bandwidth limitation of the detector B_{pd} is related to the repetition rate difference (Δf_{rep}), repetition rates (f_{r1} and f_{r2}) of the dual combs and dispersion amount ($|\beta_2|L$). They satisfy the following relationships:

$$|\beta_2|L > \frac{|\Delta f_{rep}|}{2\pi f_{r1} f_{r2} B_{pd}} \quad (\text{R1})$$

That means,

$$B_{pd} > \frac{|\Delta f_{rep}|}{2\pi f_{r1} f_{r2} |\beta_2|L} \quad (\text{R2})$$

At the dual comb and dispersion quantities used in this article, the bandwidth of the photodetector used must not be lower than 6.684 GHz. We can alleviate the bandwidth limitation by using a larger amount of dispersion. For example, when we double the current amount of dispersion, the required bandwidth will be reduced to 1/2 of the original.

(1) We compare and discuss the technical advances in table S2.

Table S2 compares key performances of recent ranging systems, including FMCW ranging, conventional dual-comb ranging, dual-comb ranging with the TPFC and the current DFT-based dual-comb ranging (this work).

Table S2. Performance comparison of distinct ranging methods

Ref	Technique	Single shot accuracy	Distance	Speed
1	FMCW Ranging	30 μm	NG	1 Hz
2	FMCW Ranging	115 μm	NG	9.6 Hz
3	EOS-TD-Based TOF	24 nm	6 mm	250 MHz
4	Microcomb DPI	5.6 μm	1.1 km	35 kHz

5	Dual-microcomb TOF	60 μm	0.15 cm	~ 1 GHz
6	Dual fiber comb TOF	3 μm	1.5 m	5.19 kHz
7	Dual Ti: Sapph comb TOF	510 μm	29 cm	220 kHz
8	Dual-comb with TPFC	125 μm	75 cm	40 Hz
9	This work	262 nm	1.7 km	85 kHz

NG: not given.

(2) We add the discussion about the CPA effect.

We add a sentence in the maintext:

Such pulse broadening is also particularly advantageous in ranging applications, especially for enhancing pulse energy through the chirped-pulse amplification (CPA) technique [47]. More results are shown in supplementary note S3.

We add related details in the supplementary note S3.

Compared to the transformation-limited femtosecond pulse, the dispersion-induced linear broadening results in a temporal width extending into tens of nanoseconds, for the pulse after DFT. This broadening allows for higher power amplification in an Erbium-Doped Fiber Amplifier (EDFA) with minimal nonlinear effects. This feature represents one of the key advantages of our method over the traditional dual-comb ranging schemes that rely on ultrashort pulses. As illustrated in **Fig. S7**, we amplify the same pulse before and after chirp-based broadening. By using the same C-band EDFA, we analyze their spectra. Due to the short duration and high peak power of transformation-limited pulses, their amplification efficiency is relatively low, making such pulses challenging to amplify effectively. For example, when the output power reaches 0 dBm, the spectrum of an ultrafast pulse undergoes significant supercontinuum broadening and deformation, with little increase in peak power. In contrast, the dispersion-stretched pulse retains its spectrum remarkably well even when amplified to 32 dBm. These results highlight the unique advantage of our DFT-based approach in enhancing pulse energy, rendering it particularly suitable for out-of-lab applications such as long-distance ranging.

Fig. S7. Amplified spectra of a pulse before and after dispersive-stretching. **a.** Amplified spectra of pulse without chirping. Curves from low to high, output power is -17 dBm, 0 dBm, 3 dBm, 10 dBm. **b.** Amplified spectra of pulse after stretching. Curves from low to high, output power is -17 dBm, 0 dBm, 3 dBm, 10 dBm, 32 dBm.

3. Specific point on the text: Abstract: P.1. “DFT-based spectrally interferometric measurement instead of pulse-fitting in time domain...” The phrase is incorrect: As follows from the reference mentioned above, dual-comb distance ranging is using TOF and interferometric technique based on processing data in a Fourier domain.

Response:

Thanks for your correction. Yes, we fully agree that dual comb ranging technique is also an interferometric measurement. Therefore, we rewrite this sentence.

The revised sentence becomes:

We demonstrate that after in-line pulse stretching, the delay of the flying pulses can be identified via the DFT-based spectral interferometry instead of temporal interferometry or pulse reconstruction.

Overall, this manuscript lacks novelty to meet criteria of Nature Communications. I recommend to reject this paper.

Response:

We appreciate your criticism and have made in-depth revisions to the manuscript. We also turn down the claim about our technical advances and refocus on the novelty in methodology. Overall, we propose a new DFT-based dual-comb ranging system, which not only extends the dual-comb ranging from the transform limited pulse to the chirped pulse, but also breaks the trade-off between precision and speed in the traditional ranging. This provides a new physical dimension for the analysis of ranging signals and enables extremely high precision without any dead zones.

Reviewer #2

In this article, the authors innovatively employed dispersive Fourier transform (DFT) in a dual-comb ranging system and demonstrated its application potentials in and out of lab setting. The combination of advanced in-pulse interferometric methodology and dual comb metrology disrupts the Nyquist sampling constraint while nullifies the trade-offs between speed and precision in dead-zone-free ranging. **Experimental results indicated impressive performances and the manuscript is generally well organized.** I strongly recommend this work to publish in Nature Communications, after the following questions were well addressed.

General response:

We are so grateful to see your encouragement on our work. We carefully address your concerns as we can.

1. The authors used fiber laser combs. In reference to integrated microcomb based ranging systems (e.g. in Chang, L. et al. Nat. Photon. 16, 95 2022; Shu, H. et al. Nature 605, 457, 2022), does the fiber combs show unique advantages in principle or technique?

Response:

Thanks for this question. We fully agree that microcomb based ranging systems have unique advantages such as high-integration and ultrafast response. But in measurement scenarios that require both long distance and high precision, fiber combs can demonstrate their merits:

- Compared with microcombs, fiber laser combs have lower repetition rate (usually in the MHz to sub-GHz range), which is conducive to dispersion broadening; at the same time, according to the TOF principle, smaller repetition frequencies help achieve larger measuring range.
- Fiber laser combs can output a higher power, and their feed-back locking system is simpler. Thus we can conveniently optimize the comb parameters.

In order to make this point clearer, we cite these two references in the text and added some explanations.

We add a sentence in our maintext:

We use fiber laser combs because their cavity parameters are flexibly editable.

2. Comb ranging methods are known for their versatility. Is the DFT applicable only in TOF? The authors should discuss whether this scheme could be implemented in other comb ranging systems, such as OFDR (e.g. in Chen, R. et al. Nature Photon. 17, 306, 2023).

Response:

Thank you for your question. Fundamentally, DFT introduces linear chirping to an initially chirp-free pulse, whereas OFDR translates frequency domain information into spatial domain insights by leveraging a linearly swept light source and analyzing the backscattering from it. Hence, we posit that the DFT approach requires minimal enhancements for integration into OFDR systems that utilize optical frequency combs. See **Fig. R4**. We propose utilizing pulses subjected to dispersion Fourier transformation in place of swept light sources. Building on this concept, we outline a straightforward experimental configuration to deploy DFT within OFDR methodologies.

Fig. R4. A simple experimental device to implement DFT-based OFDR by using fiber comb.

We mention this point in our maintext:

We further note that this approach extends the capabilities of precision measurement from pulses with transformation limits to those with chirp, thereby providing a sweeping-free effect. This advancement proves beneficial in additional metrology systems, including those based on frequency-modulated continuous waves (FMCW) and optical frequency domain reflectometry (OFDR).

3. In the maintext, the authors mainly focus on articulating their own results. To benefit a wider audience not familiar with this field, it may be helpful to include a comprehensive table that contrasts the primary indices with existing literature.

Response:

Thanks for this suggestion. We have added comparisons with metrics reported in other articles in the supplementary material so that readers can better understand the advantages of our system. In particular, in terms of distance and accuracy ratio, we demonstrate a far-leading level (6.5×10^9), which means more balanced ranging performance.

In the supplementary Note S3, we add a sub-section S3.8.

S3.8 Performance comparison

Table S2 compares key performances of recent ranging systems, including FCMW ranging,

conventional dual-comb ranging, dual-comb ranging with the TPFC and the current DFT-based dual-comb ranging (this work).

Table S2. Performance comparison of distinct ranging methods

Ref	Technique	Single shot accuracy	Distance	Speed
1	FMCW Ranging	30 μm	NG	1 Hz
2	FMCW Ranging	115 μm	NG	9.6 Hz
3	EOS-TD-Based TOF	24 nm	6 mm	250 MHz
4	Microcomb DPI	5.6 μm	1.1 km	35 kHz
5	Dual-microcomb TOF	60 μm	0.15 cm	\sim 1 GHz
6	Dual fiber comb TOF	3 μm	1.5 m	5.19 kHz
7	Dual Ti: Sapph comb TOF	510 μm	29 cm	220 kHz
8	Dual-comb with TPFC	125 μm	75 cm	40 Hz
9	This work	262 nm	1.7 km	85 kHz

NG: not given.

4. This scheme necessitates a large dispersion element (i.e., a DCF with a length of 15.2 km), which could limit its integration for broader applications. Could the authors discuss potential optimizations? Alternatively, could the authors suggest any substitutes (e.g., crystals) that might achieve a similar effect in the future?

Response:

Thank you very much and this is for sure a good question. We fully agree that kilometer-level dispersion compensation fiber will increase the volume and power loss, which is harmful to the compactness and integration of system. In the future, we can use chirped fiber Bragg grating (CFBG) to replace the long fiber. In a CFBG, since the refractive index period is chirped, different reflection wavelengths can have different delays, and large group delays can be achieved within a short length at a ratio of ns/nm. By adopting a customized CFBG that meets our needs, we will greatly improve the integration of our system in future work.

In the methods of maintext, we add two sentences:

We note that in addition to DCF, other components can also be used to provide the dispersion. For instance, customized chirped fiber Bragg gratings (CFBGs), capable of offering substantial dispersion over short distances, can be utilized to significantly enhance the system's integration.

5. In relation to the previous question, how can timing jitter in the long fiber be mitigated or suppressed in an open environment? The authors should provide experimentally measured data.

Response:

Thank you for highlighting this crucial question. In a real-world setting, the length jitter of the dispersive element (specifically, the long DCF) can affect measurement accuracy. To address this issue, we implement temperature control and active vibration isolation on the dispersion element during our experiments. This approach aims to minimize the fiber's jitter in an open environment. Specifically, we place the DCF modules in a drying chamber designed to maintain a constant temperature and humidity. These modules are set on an active vibration isolation platform to further ensure stability.

(1) In the maintext, we add more information:

The two DCF modules are positioned within a drying chamber that maintains a constant temperature and humidity environment. This chamber is situated on an active vibration isolation platform to ensure stability.

(2) In the supplementary Note S3.1, we add more measured details.

Additionally, we monitored 1,000 consecutive measurement sets (total length 2000 s) with identical pulse delays. As depicted in **Fig. S6f**, the measurement error stays within ± 1 fs. That means, the ranging error due to the long term instability is below 150 nm for a single shot. Here, the three traces show the cases that we pre-set the delays 36.95ps, 87.00ps, and 137.99ps, respectively. This indicates that by positioning the dispersive fiber in a stable environment, the error induced by its length fluctuations can be essentially mitigated.

Fig. S6. f. Statistic measurements under different pulse delays. (Pre-setting time delay: top: 36.95 ps; medium: 87.00 ps; bottom: 137.99 ps).

6. To stabilize the dual comb, the authors used two servos, this increases the complicity in system. Referring other laser stabilization schemes (e.g. J. Bowers, et al. eLight 3, 1, 2023), can the authors

discuss a way to simplify their feedback loops?

Response:

Thanks for this question. For stabilizing the repetitions, one can simplify the feedback loop in a system which generates two phase-locked laser combs simultaneously. We add the discussion in the maintext.

In the introduction, we add:

In many cases, the phase-locked dual-comb can even be generated in one cavity (either a fiber-loop or a microresonator) [34,35,36,37,38], further suppressing the common noise and simplifying the locking system.

In the supplementary note S2, we add:

In future, an alternative scheme is using a pair of counter-propagating laser combs, which share the same laser cavity, as **Fig. S3b** demonstrates. Electronic configuration is the same as single comb. In details, due to the bidirectional gain, counter-propagating dual-comb generated in a single cavity is an effective way to simplify the current dual-comb configuration. This can be realized by wavelength multiplexing, polarization multiplexing, or bidirectional mode-locking [s9, s10]. For example, **Fig. S3b** shows the experimental set-up for a bidirectional mode-locked laser comb. By fine tuning the pump power and intra-cavity polarization, frequency combs can be generated from both circulating directions, with repetition rate difference originating from center-wavelength and group-velocity dispersion. Thanks to the common-mode noise cancellation scheme, the repetition rate difference is inherently stable. Therefore, a single set of phase lock loop to lock the repetition rate of either comb is sufficient for dual-comb stabilization.

Fig. S3. Experimental setup of fiber comb generation. a, Two laser combs and their stabilization setups. **b,** Simplified design based on counter-propagating dual combs generated in one cavity. PZT: piezoelectric ceramics, PID: proportional-integral-derivative controller, LNA: low noise amplifier, LPF: low pass filter, SMF: single mode fiber, PC: polarization controller, EDF: erbium-doped-fiber, WDM: wavelength division multiplexer.

7. In dual comb ranging, reflected optical power plays a pivotal role in ensuring the Signal-to-Noise Ratio (SNR). It would be beneficial if the authors could measure this reflected power from varying distances and discuss whether an optical amplifier is necessary or not.

Response:

Thanks for the professional suggestion. We fully agree that the received optical power decreases with distance. And, types of light reflection are different in different test environments. For instance, specular reflection (e.g. using cube-corner prism) offers a much higher reflectivity than diffuse reflection (e.g. measuring an UAV).

In the supplementary note S3, we add a sub-section S3.6 to demonstrate the details and discuss whether optical amplifiers are necessary.

S3.6 Optical power and measurable distance

In dual comb ranging, the reflected optical power critically influences the Signal-to-Noise Ratio (SNR). The factors that typically constrain the received optical power include laser power, diffuse

reflection loss, the focal length and numerical aperture of the collimator, and the receiver's caliber, among others. For a LIDAR system, assessing the reflected power across different distances and evaluating the necessity of an optical amplifier are essential.

Moreover, the dominant types of reflection vary across different ranging scenarios. In measurements of spatial distance or surface morphology, specular reflection predominantly contributes to strong reflectivity. Conversely, for objects like fan blades or drones (lacking cube-corner prisms), diffuse reflection is more significant. Our tests on received optical power in these scenarios are illustrated in Fig. S11a and S11b.

Fig. S11c presents the measured received power using our dual comb ranging system to detect a UAV target, both with and without a cube-corner prism. The two curves represent the actual received optical power from a diffuse reflection target and a specular reflection target, respectively. In our experiments, we set the transmit laser power at 14 dBm (or 25.12 mW). For both diffuse and specular targets, the received power diminishes exponentially with increasing distance, primarily due to light absorption and divergence in the air. The minimum receivable power is -50 dBm, indicating that with a 14 dBm laser power, the maximum measurable distance for a diffuse object is approximately 81 m, whereas for a specular target, it exceeds 10 km.

One strategy to extend the measurable distance involves boosting the probe comb's sent-out power. Fig. S11d demonstrates this approach. Under conditions of diffuse reflection, the received optical power linearly escalates with an increase in transmitted optical power at a constant distance of 7 m. We estimate that at an optical power of 30 dBm (or 1 W) for the laser comb, our system can effectively detect a diffuse reflection target from 1 km away.

Fig. S11. Discussion of the measurable distance. a & b. Ranging scenarios, the target demonstrates diffuse reflection or specular reflection. **c.** Received ratio in power, for diffuse and specular reflection. **d.** Received optical power scales with the sent-out laser comb power.

8. The application of drone detection outside the laboratory mentioned by the authors is quite intriguing. However, I am curious about the success rate of signal reception in this context. Additionally, while the use of a corner prism to secure light reflection is appreciated, it is important to note that in real-world applications, a corner prism may not be present on every target. Therefore, the authors need to elaborate on potential solutions to this hurdle.

Response:

Thanks for your question.

1) Yes for the out-door measurement, it is difficult to guarantee perfect signal reception, as the hovering of the drone is unstable. For addressing this question, we can improve the framing rate.

2) The utilization of a cube-corner prism helps to increase the success rate of signal reception. It can boost the reflectivity and ensure the correct light path.

3) We also pretty agree that in real-world applications, a cube-corner prism may not be present on every target. Therefore we discuss this point in the revised supplementary Fig. S11. To address the problem that reflectivity is low, we can:

- Increase the laser power by using an optical amplifier. The stretched pulse is suitable for further amplification.
- Select an optical transmitter and an optical receiver with larger numerical aperture.
- Use photoelectric detection equipment with higher sensitivity.

We add the related information in the maintext:

Alternatively, one can increase the output power of the probe comb via amplification or use a receiver with a larger numerical aperture.

9. The textual elements in some figures within the article seem inconsistent, and there are typographical errors scattered throughout. I would suggest that the authors diligently refine their manuscript before submission for acceptance.

Response:

Thank you for your kind reminder. We have reviewed the full text and made changes to the format and syntax.

In sum, this research presents an innovative physical mechanism for rapid and precise LIDAR signal

demodulation. The authors offer substantial results that validate the applicability of this method, both within and outside the lab confines. Given its timeliness and broad appeal, I would recommend publication in Nature Communications, contingent upon the authors' willingness to make major revisions to their manuscript.

Response:

Thank you again for your recommendation.

Reviewer #3

The paper “Dispersive Fourier transform based dual-comb ranging”, by Chang et al., reports on the implementation of a dual frequency comb system for optical ranging. Specifically, the authors leverage fibered dispersive Fourier transform for time-stretching the frequency combs. Instead of using time-of-flight measurement approaches for optical ranging, the proposed system takes advantage of an interferometric technique to retrieve absolute distance measurements.

In this paper, the implemented system takes advantage of a particular approach combining asynchronous dual comb measurements based on two soliton phase locked lasers to provide the required Vernier effect, along with dispersive Fourier transform to achieve temporal encoding of the comb spectra in the temporal domain. The latter allows for obtaining a time-dependent interferometry between the different combs (local-reference and local-probe) that are used to achieve efficient distance measurements with nanoscale resolution. **The proposed technique allows for extended non-ambiguity range and drastically reduced dead-zone that are common issues in most LIDAR ranging techniques. The results are convincing and proof-of-concept demonstrations are illustrated via two practical examples (for out of the lab ranging, and high-resolution dynamic morphology measurements).**

Overall, the paper is scientifically sound and innovative enough to be considered for publication in Nature Communications. I think that the proof-of-concept demonstrations are significant, and the overall approach is promising. I however believe that several points, appended below, should be addressed before considering the manuscript for publication.

General response:

We sincerely thank you for your positive comments and are grateful to see your suggestions. Here, we address your questions one by one as below.

1. I appreciated the overall scientific quality of the paper but the writing could be reasonably improved by thorough proofreading.

1a/First, several sentences are maybe overselling/general and do not bring much while being present at the expense of the overall clarity of the sentence. I would recommend rephrasing these for the understanding of the reader. Some examples:

i) Line 43: Recently, the comb stabilization technology has pushed the light towards the level of “photonic clock”...

ii) Line 249: the DFT-based dual comb LIDAR scrutinizes Fraunhofer’s diffraction with exceptional resolution, shattering the Nyquist sampling constraint while nullifying the trade-off

between speed and precision in ranging.

Response:

i) Thanks. In the introduction. We rewrite this sentence [line 43 in previous version] as:

Recently, the development of comb stabilization technology [10,11], further enables ultrahigh accuracy for LIDARs [12,13,14,15].

ii) In the conclusion, we rewrite this sentence [line 249 in previous version] as:

Different from conventional methods reliant on temporal pulse separation retrieval, the DFT-based dual-comb LIDAR precisely examines Fraunhofer diffraction in spectrum, thus provides a solution to overcome the compromise between speed and accuracy in dual-comb ranging.

1b/ Some wordings also appear poorly chosen and requires the reader to look into/guess what was meant. This should be addressed for the scientific clarity of the paper. Some examples:

i) Line 43: integral?

ii) Line 54: suppresses => suppressing

iii) Line 74: transforms the time-of-flight based ranging mechanism from temporal separation to spectral interferogram, and thus overcomes the trade-off between speed and accuracy (understandable, but unclear)

iv) Line 76: Analyzing

v) Line 84: swapping => sweeping

vi) Line 98: T1 and T2 are not really defined in the main text

vii) Line 190: In our scheme, the minimum detectable τ is mainly determined by the frequency resolution of the Fourier transform peak (hundreds of nanometers) => What does hundreds of nanometers refer to?

Response:

Thank you so much for your careful reading. We check the points one-by-one and reinforce our statements.

i) The “integral” is replaced to: **critical**.

ii) We correct the tense to: **suppressing**.

iii) We rewrite the sentence to: **By measuring intra-pulse interference rather than temporal separation, this approach surpasses the sampling limitation in conventional dual comb rangefinders, and thus overcomes the trade-off between speed and accuracy.**

iv) Line 76: We change “analyzing” into “**analysis**”.

v) Line 84: As mentioned in ref. [24]: "To greatly extend this range ambiguity, we performed a similar distance measurement after manually swapping the roles of two soliton streams". What we

want to say here is that by exchanging the roles of the two combs, the extension of the ambiguous range can be achieved. **So the use of swapping here is correct.** However, to avoid misunderstanding, we modify this sentence as follows: **exchanging the roles of the two combs with repetition rates f_{r1} and f_{r2} [18]**

vi) Line 98: We are sorry for the confusion. We add the definitions of T_1 and T_2 now: **When using two frequency combs with distinct repetition periods T_1 and T_2**

vii) Line 190: we make this statement more clearly: **In our scheme, the minimum measurable distance is just determined by the minimum detectable τ , which is < 1 fs, as **Fig. 2** displayed.**

2. Figure 1 b is heavy and can be unclear to the reader (what does τ_i refer to ?). I think integrating the local reference overlap in blue and explaining things a bit better may provide further clarity to the reader (how is the time offset measured). Maybe integrating Fig. S1 in the main text could be useful?

Response:

Thank you for this suggestion. In the revised manuscript, we reorganize the **Fig. 1b** and add new explanations, hoping it could be clearer.

Specifically, pulse-to-pulse offsets between the reference comb and the local comb could be τ_1 (between pulse i & iii) and τ_2 (between pulse ii & iv), while pulse-to-pulse offsets between the probe comb and the local comb could be τ_3 (between pulse v & vii) and τ_4 (between pulse vi & viii), here $\tau_1 + \tau_2 = \tau_3 + \tau_4 = \Delta T$.

Now we explain our strategy. The left green box shows the region that the reference pulses overlap with the local pulses, while the right yellow box shows the region that the returned probe pulses (with a TOF delay) overlap with the local pulses, the accurate TOF $\Delta\tau = T_1 - N(T_1 - T_2) + \tau_1 - \tau_3$, or $\Delta\tau = T_2 - N(T_1 - T_2) + \tau_2 - \tau_4$.

Fig. 1. Conceptual design and operation of the DFT based dual comb ranging. b, Mechanism of

the high precision time delay analysis using DFT. Slight mismatches of the pulses are tested via spectral interference.

We also note, this operation is developed from dispersion interference, so we add a reference in our introduction:

[19] Jang, Y.-S. *et al.* Nanometric Precision Distance Metrology via Hybrid Spectrally Resolved and Homodyne Interferometry in a Single Soliton Frequency Microcomb. *Phys. Rev. Lett.* **126**, 023903 (2021).

3. The real scale of Fig 2c is missing and the zoom does not allow identifying the beat note bandwidths. It is therefore hard to understand where/how the 3 MHz bandwidth and below 90 kHz resolution are obtained. Can we get an idea of the SNR of the beat note vs the noise floor or the potential presence of other frequency components in the RF spectrum?

Response:

Thanks for this question. In the revised manuscript, we reorganize the **Fig. 2**, with reinforced description:

More in details, we zoom one of the Fourier transformed peaks ($f_{i,1}$) in **Fig. 2d** for example. Thanks to the high stability of the interference, linewidth of each Fourier peak is < 3 MHz, and typical signal-to-noise ratio (SNR) of each frequency components is > 30 dB.

Fig. 2. Single point ranging. d, Zoomed-in spectrum of the $f_{i,1}$ peak, with 3-dB bandwidth 2.9 MHz and SNR 32 dB.

4. In the supplementary, I believe that the authors may explain further how the ambiguity range is extended by “exchanging” the role of the combs as illustrated in Fig S9.

Response:

Thanks for this note. In the revised supplementary notes S3.4, we add the theoretical derivation that exchanging double combs can extend the ambiguity range:

The non-ambiguity of a dual-comb ranging system can be significantly extended by leveraging the Vernier effect. This is achieved through the manual interchange of the roles between the two combs. In this discussion, we delve into the mechanism behind this. Consider the repetition rate of the signal comb is f_{r1} and the repetition rate of this local comb is f_{r2} , a distance can be written as

$$L = \frac{c}{2}(MT_1 + \Delta t_1) \quad (24)$$

Here $T_1 = 1/f_{r1}$ is the period of the comb #1. M is an integer. If $M=0$, L is in the non-ambiguity region of comb#1. However, when the repetition rate of the signal comb is f_{r2} , and the repetition rate of this local comb is f_{r1} , this distance is

$$L = \frac{c}{2}(NT_2 + \Delta t_2) \quad (25)$$

Here $T_2 = 1/f_{r2}$ is the period of the comb #2. N is an integer. If $n=0$, L is in the non-ambiguity region of comb#2. In above equations, we call Δt_1 and Δt_2 ‘wrapped times’, $\Delta t_1 \leq T_1$, $\Delta t_2 \leq T_2$. For a given distance L , $MT_1 + \Delta t_1 = NT_2 + \Delta t_2$. Commonly in a dual comb ranging system, difference between T_1 and T_2 is tiny, for measuring a fixed L , $M = N$ [s11]. Therefore, we can obtain:

$$M = \frac{\Delta t_2 - \Delta t_1}{T_1 - T_2} \quad (26)$$

Therefore, the given distance L can also be written as:

$$L = \frac{c}{2} \left(\frac{\Delta t_2 - \Delta t_1}{T_1 - T_2} T_1 + \Delta t_1 \right) = \frac{c}{2} \left(\frac{\Delta t_2 T_1 - \Delta t_1 T_2}{T_1 - T_2} \right) \quad (27)$$

We can regard this equation as a binary function $L(\Delta t_1, \Delta t_2)$. When $\Delta t_2 = T_2$ and $\Delta t_1 = 0$, L reaches the maximum value:

$$L_{\max} = \frac{c}{2} \left(\frac{T_2 T_1}{T_1 - T_2} \right) = \frac{c}{2\Delta f_{rep}} \quad (28)$$

Relatively, when using using one single comb, one can easily know the maximum measurable distance is $c/2 f_{rep}$. Typically, $\Delta f_{rep} \ll f_{rep}$, so that exchanging the roles of dual comb can extend the maximum measurable distance. Correspondingly, the use of double comb swapping operation will also slow down the frame rate.

5. Fig.1d-e needs additional information to be properly assessed and can be completed based on Fig S2.

Response:

Thanks for this suggestion. Referring the **Fig. S2**, we enrich the information in the revised **Fig. 1**. For better clarification, we change the panel order in the **Fig. 1**. Now the calculations are discussed in

Fig. 1e-1f.

Fig. 1e and **1f** discuss the calculated parametric spaces. In **Fig. 1e**, we show the δ linearly increases with $|\beta_2|L\Delta\lambda$, with a fixed $\lambda = 1550$ nm. In our DFT system, δ must be smaller than repetition period of the probe comb (40.87 ns), as the dashed red curve marks. Here the white dot presents the selected parameter in our experiment. In **Fig. 1f**, we map the relationship between the $|\beta_2|L$, the τ and the f_i . On one hand, f_i is proportional to τ , with a coefficient $|\beta_2|L$. In experiment, the maximum detectable f_i is determined by the bandwidths of the oscilloscope ($B_o = 20$ GHz), as the yellow dashed line marks. On the other hand, we must ensure that the detectable τ should be larger than the period difference $|T_1 - T_2|$ (141.5 ps), the red dashed line shows this requirement. Hence, the minimum $|\beta_2|L$ is $\Delta f_{rep}/(2\pi f_{r1} f_{r2} B_o) = 1127$ ps², as the green solid line illustrates. In our experiment, we set $|\beta_2|L = 3371.47$ ps² (white solid line), locates in the working region.

Fig. 1. Conceptual design and operation of the DFT based dual comb ranging. e & f, parametric spaces. Both the stretching ratio and the f_i are determined by the group-delay dispersion $|\beta_2|L$.

6. More generally, the supplementary document contains essential information regarding the dead zone management based on experimental parameters (chromatic dispersion, PD bandwidth, repetition rate). Therefore, some information contained on S1.3 should be recalled in the main document, for the reader to properly estimate the related limitations.

Response:

Thanks for this suggestion. In the section discussing the dead zone, we recall the experimental parameters for readers:

In **Fig. 3e**, we explore the parametric boundaries necessary for eliminating dead zones. The unmeasurable area length, L_{dead} , in DFT-based demodulation is influenced by the comb repetition intervals T_1 and T_2 , the photodetector's bandwidth B_{pd} , and the total dispersion $|\beta_2|L$, as outlined in the equation:

$$L_{dead} = \frac{c}{2} \left[T_1 - \frac{4\pi T_1 |\beta_2| L B_{pd}}{T_1 - T_2} \right] \quad (4)$$

It is evident that an increase in $|\beta_2|L$ results in a decrease in L_{dead} . Achieving L_{dead} signifies that the system is free of dead zones, as red dashed curve shows. In our experiments, with $B_{pd} = 25$ GHz, $T_1 = 40.87$ ns, $T_2 = 40.73$ ns, the employment of a $|\beta_2|L = 3371.47$ ps² is significantly above the requirement for achieving a ‘dead zone free’ state (white dot).

Fig. 3. Elimination of dead zones. e, Parametric limitation for reaching the dead zone free measurement.

7. Can the authors complete the experimental set-up description in the main document regarding the device capacity and long term stability, two criteria of main importance for embedded systems.

Response:

Thanks for mentioning this point.

In the maintext, we mention this point:

Based on technological optimization, our DFT based dual comb ranging system exhibits modular capability and long-term reliability, enabling application validation both inside and outside the laboratory (supplementary note S3).

In supplementary note S3, we add a section to clarify that our system could be compact, compatible with electronic systems, and have acceptable long-term stability.

S3.7 System capacity and reliability

In this discussion, we address the aspects of capacity, compatibility, and long-term stability of our devices, which are significant for the functioning of embedded systems. Initially, we introduce a dual comb module based on mode-locking fiber laser technology. This module can be efficiently packaged in a compact box within $20 \times 15 \times 10$ cm³ and weighing less than 1 kg, as illustrated in **Fig. S12a**. Within this module, we have seamlessly integrated a power supply, a 980 nm pump laser, two sets of mode-locked lasers, and two feedback stabilizers. The internal architecture is showcased in **Fig. S12b**, displaying the thoughtful separation between the optical and electrical components. The design incorporates two output ports intended for emitting two laser combs with distinct repetition rates.

Concurrently, the system benefits from temperature control and vibration reduction features. Thanks to a vibration isolation treatment, each fiber laser comb can be easily connected and used (plug and play) with good stability. As a result, intensity uncertainty of each laser comb is maintained below 0.1% over a 48-hour period (as shown in **Fig. S12c**).

Fig. S12d reveals the prototype of our ranging system. This system extends beyond the laser comb source module to include a Pan-tilt, a transmitter & receiver, a signal processor, and a dispersive delay component. All components are designed to fit within a platform having overall dimensions of $60 \times 40 \times 30 \text{ cm}^3$. To effectively visualize the outcomes of dual comb ranging, the utilization of an external oscilloscope is indispensable. **In Fig. S12e**, we demonstrate the scenario of data collection using an oscilloscope in a single frame. Concluding our assessment, we examine the long-term reliability of our ranging system. Conducted in a laboratory setting, we target a static object (mirror, 3.4643 m away from the transmitter) for a continuous duration of 12 hours, during which our system diligently records the measured distance data (illustrated in **Fig. S12f**). This testing confirms that the ranging error remains below 10^{-5} . We identify that the system's inherent uncertainty stands at $\pm 300 \text{ nm}$, while the target fluctuation is recorded at $\pm 2.5 \text{ }\mu\text{m}$.

Fig. S12. Capacity and reliability of the system. **a.** Packaged device of our mode locked fiber laser based dual comb ($20 \times 15 \times 10 \text{ cm}^3$). **b.** Internal architecture of the dual comb device. **c.** Long-term stability of the laser comb output. **d.** Picture of the DFT based dual-comb ranging system prototype. **e.** Data collection in an oscilloscope. **f.** Long-term stability for ranging a fixed target.

8. Can the authors comment and discuss the sensitivity of this DFT-based ranging method? As the femtosecond pulse is temporally stretched, the power temporal (DFT spectral) density is fairly weak. As the diffuse reflection power is -17 dBm , and the DFT temporal bandwidth is of the same order of

magnitude as the comb period, the peak power is almost equal to the average power and one would thus expect a visibility measurement based on $\sim 100 \mu\text{W}$ probe peak power detection (at best). What is the impact on the sensitivity in this case? What is the sensitivity threshold for measurable fringe visibility with sufficient SNR for suitable ranging? Any idea of the maximal achievable ranging length considering this decreased peak power considering absorption and beam divergence?

Response:

Thanks for this question.

(1) About sensitivity of our ranging system: We add more information, demonstrating that for specular reflection, the sensitivity is almost 100%; for diffuse reflection, the sensitivity is -26.5 dB. The sensitivity is majorly limited by the target reflectivity, the laser power, and the receiving efficiency. Noise base of our photodetector is about -50 dBm, while fast RIN base of the system is about -90 dBm.

(2) About the transmitted power and the received power: We add more discussion, showing that the maximum output laser power can be 2 W (33 dBm), after amplification. We note that after stretching operation, the pulses are better suitable for further amplification. In this case, when a target (diffuse reflection) locates at 10 m away, the received power reaches $10 \mu\text{W}$ (-20 dBm).

(3) About the maximal achievable ranging length. We add more measurements, illustrating that for a target with specular reflection, the maximal achievable ranging length is $> 1 \text{ km}$ (without high-power amplification), while for a target with diffuse reflection, the maximal achievable ranging length is 81 m (without high-power amplification).

In the maintext, we add:

Alternatively, one can increase the output power of the probe comb via amplification or use a receiver with a larger numerical aperture.

In the supplementary note S3, we add a sub-section S3.6 to discuss the sensitivity and the achievable ranging distance.

S3.6 Optical power and measurable distance

In dual comb ranging, the reflected optical power critically influences the Signal-to-Noise Ratio (SNR). The factors that typically constrain the received optical power include laser power, diffuse reflection loss, the focal length and numerical aperture of the collimator, and the receiver's caliber, among others. For a LIDAR system, assessing the reflected power across different distances and evaluating the necessity of an optical amplifier are essential.

Moreover, the dominant types of reflection vary across different ranging scenarios. In measurements of spatial distance or surface morphology, specular reflection predominantly contributes to strong reflectivity. Conversely, for objects like fan blades or drones (lacking cube-corner prisms),

diffuse reflection is more significant. Our tests on received optical power in these scenarios are illustrated in **Fig. S11a** and **S11b**.

Fig. S11c presents the measured received power ratio using our dual comb ranging system to detect a UAV target, both with and without a cube-corner prism. The two curves represent the actual received optical power from a diffuse reflection target and a specular reflection target, respectively. In our experiments, we set the transmit laser power at 14 dBm (or 25.12 mW). For both diffuse and specular targets, the received power diminishes exponentially with increasing distance, primarily due to light absorption and divergence in the air. The minimum receivable power is -50 dBm, indicating that with a 14 dBm laser power, the maximum measurable distance for a diffuse object is approximately 81 m, whereas for a specular target, it exceeds 10 km.

One strategy to extend the measurable distance involves boosting the probe comb's sent-out power. **Fig. S11d** demonstrates this approach. Under conditions of diffuse reflection, the received optical power linearly escalates with an increase in transmitted optical power at a constant distance of 7 m. We estimate that at an optical power of 30 dBm (or 1 W) for the laser comb, our system can effectively detect a diffuse reflection target from 1 km away.

Fig. S11. Discussion of the measurable distance. a & b. Ranging scenarios, the target demonstrates diffuse reflection or specular reflection. **c.** Received ratio in power, for diffuse and specular reflection. **d.** Received optical power scales with the sent-out laser comb power.

REVIEWERS' COMMENTS

Reviewer #1 (Remarks to the Author):

The authors provided detailed response to the points raised in the reports. I think the revised version presents results and claims in a more accurate way. I would thin that the revised paper can be recommended for publication in the Nature Communications and it will be of interest for community.

Reviewer #2 (Remarks to the Author):

I had reviewed this manuscript submitted to Nature Communications. Authors have replied to my previous comments and modified the manuscript. The quality of this version is significantly better. The exploration is original and meaningful and the topic is of interest to readers in related disciplines. I think this paper can be published in Nature Communications.

Reviewer #3 (Remarks to the Author):

The authors provided a careful and substantial iteration on the manuscript, clearly putting forward the advantages and limitations of their proposed system. They have adressed all my previous concerns (Reviewer #3. I am thus glad to support the publication of their work in Nature Communications.